# Kidney disease impairs tendon function in rats

Christopher M. T. Hayden[1,2], Natalie K. Gilmore[1,2] (iD), Benjamin Osipov[3], Sarah E. Brashear[2] (iD), Marc Gorge[2], Kuang-Yu Jen[4] (iD), Lucas R. Smith[2] (iD), Blaine A. Christiansen[3], Baback Roshanravan[5] and Keith Baar[1,2,6] (iD)

[1]*Molecular Cellular and Integrative Physiology Graduate Group, University of California Davis, Davis, California, USA*
[2]*Department of Neurobiology, Physiology and Behavior, University of California, Davis, Davis, California, USA*
[3]*Department of Orthopaedic Surgery, University of California Davis Health, Sacramento, California, USA*
[4]*Department of Pathology and Laboratory Medicine, University of California Davis Health, Sacramento, California, USA*
[5]*Department of Medicine, Division of Nephrology, University of California Davis HealthCalifornia, Sacramento, USA*
[6]*Department of Physiology and Membrane Biology, University of California DavisCalifornia, Davis, USA*

Handling Editors: Bettina Mittendorfer & Hirotaka Iijima

The peer review history is available in the Supporting Information section of this article (https://doi.org/10.1113/JP289753#support-information-section).

**Chris M. T. Hayden** is a PhD candidate at the University of California Davis under the guidance of Dr Keith Baar. Originally hailing from the great state of Maine, Chris received an M.S. in the Muscle Physiology Lab at UMass Amherst and a B.S. in exercise science from Springfield College. His research interests include cardio–kidney–metabolic syndrome, its effects on the musculoskeletal system and the development of interventions using an integrative multisystem approach.

[Correction made on 18 February 2026, after first online publication: Author name Juang-Yu Jen has been corrected to read "Kuang-Yu Jen."]

The Journal of Physiology

**Abstract figure legend** This study demonstrates for the first time that tendon strength is reduced in a rodent model of chronic kidney disease (8 weeks of 0.25% adenine feeding) and confirms concurrent dysfunction in muscle and bone. These findings provide novel characterization of multiple tissues, paving the way for future investigations into the effects of CKD on tendon and integrated musculoskeletal health.

**Abstract**  Spontaneous tendon rupture occurs in a concerning number of individuals with chronic kidney disease (CKD); however almost no data exist regarding CKD-related tendon pathology. Given that tendon ruptures have a significant impact on health and well-being we sought to determine whether tendon mechanics are altered by kidney disease in an established rat model of CKD. Male and female Sprague–Dawley rats (age = 8 weeks) were equally divided into a control group (CON, $n = 16$) and a group fed a diet containing 0.25% adenine to induce kidney disease ($_{ADI}$CKD). After 9 weeks, Achilles and tibialis anterior (TA) tendons were excised, and maximum tensile load (MTL), failure stress, modulus and cross-sectional area (CSA) were measured and evaluated by two-way ANOVA (main effects: CKD and sex). CKD was confirmed through elevated creatinine (1.99 *vs.* 0.61 mg/dl, CKD *vs.* CON, $P < 0.001$) and blood urea nitrogen (93.4 *vs.* 21.4 mg/dl, $P < 0.001$). Plantar flexor strength was 13% lower in $_{ADI}$CKD ($P = 0.0214$), and femur yield force was decreased by 41% in male $_{ADI}$CKD ($P < 0.001$). The failure stress of TA tendons was 24% lower in CKD *vs.* CON ($P = 0.0383$). There were no statistically significant differences in TA tendon MTL, modulus or CSA. There were no significant main effects for any parameter for the Achilles; however, *post hoc* testing following a finding of group-by-sex interactions revealed that in females Achilles failure stress was decreased in $_{ADI}$CKD by 25% ($P = 0.0283$). To our knowledge this is the first direct evidence that tendon weakness is caused by kidney disease, providing a model for further evaluating mechanisms and interventions.

(Received 22 July 2025; accepted after revision 3 December 2025; first published online 29 December 2025)
**Corresponding author** K. Baar: One Shields Ave, 195 Briggs Hall, Davis, CA 95616, USA.    Email: kbaar@ucdavis.edu

**Key points**

- Chronic kidney disease is well known to lead to musculoskeletal dysfunction, such as bone and muscle wasting.
- Numerous case reports suggest that chronic kidney disease may also predispose patients to catastrophic tendon injuries; however, there are no mechanistic studies or animal models addressing this phenomenon – and thus, no treatments.
- We determined whether a rat model of chronic kidney disease previously used to study muscle and bone dysfunction (0.25% adenine in the diet) would also cause impaired tendon function.
- Eight weeks of adenine-induced kidney disease caused tibialis anterior tendons to tear at 24% lower stress than in control animals, while also decreasing plantar flexor muscle strength by 13% and femur strength by 41% (males only).
- This is the first experimental evidence for a direct effect of chronic kidney disease on tendon function, establishing a comprehensive multitissue model for future research.

## Introduction

Chronic kidney disease (CKD) – which affects one in seven US adults (Centers for Disease Control & Prevention, 2023) – commonly leads to musculoskeletal dysfunction, including muscle wasting (Chatzipetrou et al., 2022) and bone disorders (Moe et al., 2006). This muscle and bone dysfunction along with the consequent mobility impairment and decreased physical activity are highly predictive of mortality in the CKD population (Chatzipetrou et al., 2022; Roshanravan et al., 2013) and thus has been extensively researched (Damasiewicz & Nickolas, 2018; Price et al., 2022). The musculoskeletal system, however, consists of multiple other tissues – such as ligaments, tendons and cartilage – that are necessary to provide structure, maintain posture and enable movement. These musculoskeletal tissues may also be negatively affected in CKD (Centers for Disease

Control & Prevention, 2023). Tendons, for instance, have received relatively little attention in the CKD literature, which is concerning, as there are more than 60 years of case studies describing spontaneous (non-traumatic) tendon ruptures in patients with CKD (Allata et al., 2023; Basic-Jukic et al., 2009; Fakru et al., 2021; Preston, 1972; Shah, 2002; Taşoğlu et al., 2016; Tsourvakas et al., 2004; Üreten et al., 2008; Wilson, 1957; Zaidenberg et al., 2015; Zhang et al., 2020). These reported ruptures occur in both load-bearing (Achilles and quadriceps) and non-weight-bearing tendons (biceps brachii, triceps), often during normal daily activities such as walking downstairs. The site of rupture varies across case reports and may occur at the osteotendinous junction, myotendinous junction or the tendon mid-substance (Lotem et al., 1978; Shah, 2002).

Spontaneous tendon ruptures are debilitating injuries that commonly require surgery, costing the patient more than $4000 on average in healthcare charges (Erickson et al., 2014) and a similar amount in indirect costs from missed work (Nilsson et al., 2020). These injuries lead to decrements in muscle strength, endurance and physical function that can take more than a year to fully recover from (Eliasson et al., 2018). Furthermore in haemodialysis patients tendon rupture is associated with an increased risk of hip fracture (Pichone et al., 2024), which has a generally poor prognosis. Between 1999 and 2013 1091 cases of Achilles tendon rupture were identified among Medicare patients with end-stage renal disease (ESRD), with an incidence rate of 3.8/10,000 person-years (Humbyrd et al., 2018). This is three- to fourfold greater than the rate found in a separate study across all Medicare patients (0.67–1.08/10,000 patients per year from 2005 to 2011) (Erickson et al., 2014). Incidence rate further increases to 5–6/10,000 person-years when including only transplant waitlist and recipient patients (Humbyrd et al., 2018). Furthermore although CKD patients constitute 10% of the world population, they make up 37% of reported cases of bilateral spontaneous quadriceps tendon rupture, with nearly half of these occurring prior to the initiation of dialysis (Shah, 2002). Despite these facts, the impact of CKD on tendon health is virtually unstudied, and purported mechanisms have not been experimentally investigated.

Although no concrete link between CKD and tendon rupture has been established, evidence suggests that spontaneous rupture does not occur in healthy tendons (Kannus et al., 1991). Therefore CKD or its treatments likely leads to degenerative changes in tendon tissue that predispose patients to catastrophic injury. Such degenerative changes likely contribute to chronic musculoskeletal pain, which limits movement and impairs quality of life in ∼40% of CKD patients (Caravaca et al., 2016). To date most scientific literature on the topic of tendon dysfunction in CKD consists of case studies, which provide limited insight into the mechanistic causes.

Due to the scarcity of non-invasive techniques for interrogating tendon function and the necessity of biological samples for mechanistic research, animal models are needed to understand the impact of CKD on tendons and to test interventions targeted at improving tendon function. Therefore the goal of this study was to evaluate the use of the adenine-diet-induced CKD model ($_{\text{ADI}}$CKD) in rats for studying CKD-related tendon dysfunction. The $_{\text{ADI}}$CKD model of kidney disease was first used in rats by Yokozawa et al. in 1986 (Yokozawa et al., 1986) and later refined in both rats (Diwan et al., 2013) and mice (Jia et al., 2013). This model was chosen because it has been well characterized and recapitulates the human CKD phenotype, including cardiovascular (Diwan et al., 2013, 2014) and bone (Ogirima et al., 2006; Saito et al., 2021) abnormalities. Furthermore as opposed to the popular 5/6 nephrectomy model, the adenine diet model does not require a surgical intervention (kidney ligation), has a better survival rate, and was identified in mice as the superior model for studying skeletal myopathy in CKD (Kim et al., 2021). Because tendons work in conjunction with muscle and bone to enable musculoskeletal function, we studied the effect of $_{\text{ADI}}$CKD on these tissues as well.

## Methods

### Ethical approval

All animal experiments were approved by the University of California Davis Institutional Animal Care and Use Committee under protocol #22806. The UC Davis IACUC is regulated by the Association for Assessment and Accreditation of Laboratory Animal Care International (#000029), NIH Public Health Service (assurance number: A3433-01) and US Department of Agriculture (registration number: 93-R-0433). All efforts to minimize animal pain and suffering were performed in accordance with the University's Teaching and Research Animal Care Services protocols. All research complied with the policies set forth by *The Journal of Physiology* and the ARRIVE guidelines 2.0 (Percie Du Sert et al., 2020).

### Animal model

Thirty-two Sprague–Dawley rats (age 7 weeks, $n = 16$ male and female) were obtained from Charles River Laboratories (Wilmington, MA, USA). Rats were housed two per cage with 12 h light/dark cycles and *ad libitum* access to food and water. After acclimation, animals were split into control or $_{\text{ADI}}$CKD groups such that mean maximal running speed from an exercise tolerance test

was not different between groups (Appendix A1). Animals were kept on standard chow (AIN 93G, 0.3% phosphorus, 0.5% calcium, Bio Serv F3156, Flemington, NJ, USA) until 8 weeks of age, at which point $_{ADI}$CKD animals were switched to an adenine-containing chow (0.25% adenine, 0.9% phosphorus, 0.6% Ca, Inotiv TD.180493, West Lafayette, IN, USA). These diets were maintained for 9 weeks until sacrifice, with exercise performance testing completed prior to diet initiation and after 8 weeks on the diet (Fig. 1). All animals had *ad libitum* access to food and water. Animals and food were weighed once per week. Weekly change in food weight was divided by the number of rats in the cage to estimate consumption of each rat (Fig. 1).

### *In vivo* muscle testing

Two days after the exercise tolerance test *in vivo* muscle function was tested. Rats were anaesthetized in a sealed chamber with 2% isoflurane inhalant. Once the absence of a righting response was confirmed and breathing rate had slowed they were transferred to the force transducer platform, and anaesthesia was maintained using an anaesthetic nose cone. Adequate sedation was confirmed by a lack of pedal withdrawal reflex. Fur

was then removed from the right hindlimb using an electric razor. Rats were placed in front of the force transducer (Aurora Scientific, Aurora, Ontario) with their knee positioned in a custom three-dimensional (3-D)-printed brace (experimental set-up, Fig. 4A). The hip, knee and ankle of the tested leg were aligned at 90 degrees, and the foot was taped to the force transducer pedal. Forceps were then used to pull the skin of the right hindlimb away from the distal third of the gastrocnemius muscle, and a needle electrode (negative) was then inserted into the skin without puncturing the muscle. The needle electrode was positioned such that when the skin was released, it pulled the electrode up against the gastrocnemius just distal to the muscle belly. A second needle electrode (positive) was then placed roughly 1–2 mm distal to the first in the same manner. Tape was wrapped around the needles approximately 2–3 mm from the tips of the needles to ensure consistent separation between electrodes. Electrodes were placed distally to avoid stimulation of the hamstring muscles. The stimulation box was set to 30 mA and biphasic pulse. Next, the computer was programmed to deliver one 150 Hz stimulation every other second (InstantStim), and electrode placement was verified by the presence of a muscle twitch and force tracing. If the force tracing had signs of co-contraction,

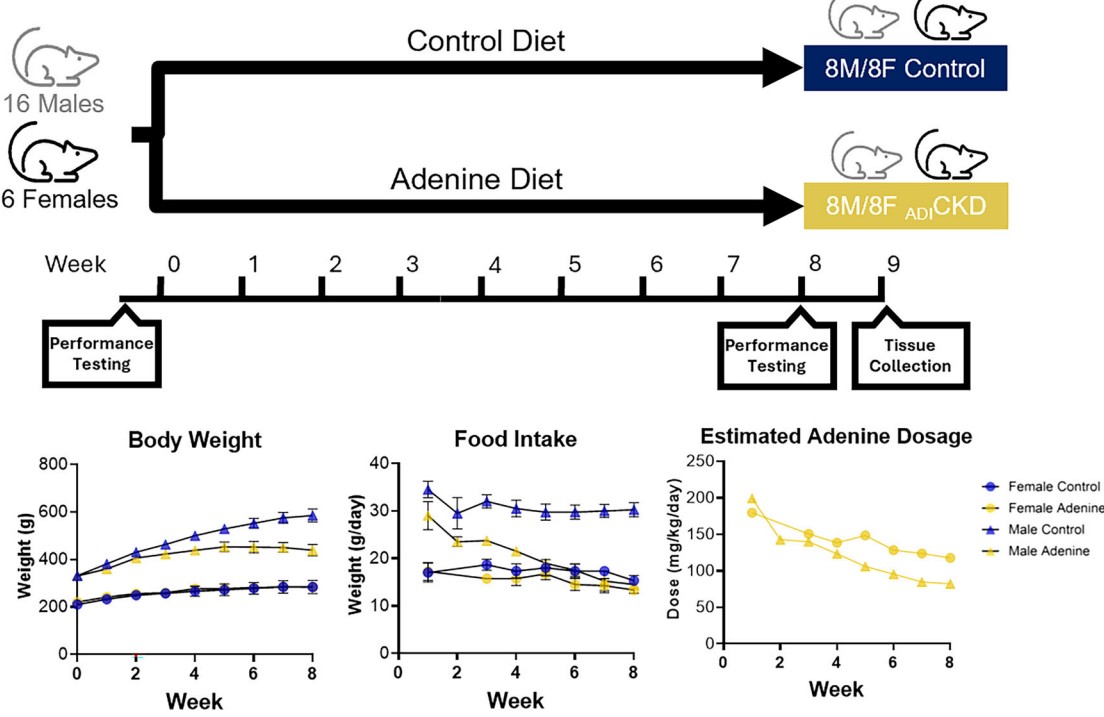

**Figure 1. Study overview and time course of body weight, food intake and estimated adenine dosage in rats fed control or adenine-containing diets**
Performance testing consisted of treadmill running and electrically stimulated muscle contraction for assessment of strength and endurance of plantar flexors (*n* = 8 per group). Standard deviation is omitted where bars are smaller than symbol size. Standard deviation could not be accurately calculated for adenine dosage and as such should be considered an estimation. $_{ADI}$CKD, adenine-diet-induced chronic kidney disease.

such as double peaks or negative force deflections, then electrodes were repositioned.

Current was set at 30 mA for all experiments to keep the stimulus consistent. This value was chosen to elicit a peak twitch force without evidence of hamstring, quadricep or dorsiflexor activation, such as changes in body position or negative deflections in the force tracing. A force-frequency test was performed by stimulating at 20, 40, 60, 80, 100, 125, 150 and 200 Hz (pulse width 0.2 ms, duration 0.350 s) with 20 s of rest between contractions. Torque traces were evaluated to ensure that tetanus was achieved and that a torque plateau had been reached at higher frequencies. Following the test, data were plotted as force *vs.* frequency to verify that a plateau had been reached, and the peak force was selected. Torque (mN•mm) was calculated by multiplying the force reported by the system (mN) by the lever arm length used during calibration (32 mm). The muscle endurance protocol was performed following a 3-min rest period after maximal force testing and consisted of 180 stimulations at 70 Hz (pulse width 0.2 ms, duration 0.350 s) with 2 s between contractions, which has previously been used to evaluate muscle endurance in 5/6 nephrectomized rats (Fusagawa et al., 2023).

### Killing and tissue collection

In the ninth week of diet intervention rats were anaesthetized with 2.5% isoflurane inhalant, and adequate sedation was confirmed by a lack of pedal withdrawal reflex. Tissues were collected immediately before killing via pneumothorax and cardiac puncture. Soleus muscles and right kidneys were excised, snap frozen in liquid nitrogen and stored at −80°C. Left kidneys were excised and immersed in 1.5 ml of 4% paraformaldehyde for 7 days at 4°C before being transferred to 70% ethanol and sent for histology. Left and right Achilles and tibialis anterior (TA) tendons were isolated along with the foot and stored in phosphate-buffered saline (PBS)–soaked gauze at −30°C. The left and right femur and humerus bones were excised, half were stored in PBS–soaked gauze at −80°C for biochemistry and the other half stored in 70% ethanol for micro-computed tomography (micro-CT). Diaphragm muscle was isolated along with several ribs, placed in Ringer's solution and immediately mechanically tested. Blood was collected in heparin-coated syringes via cardiac puncture, and ∼100 μl was immediately tested using an i-STAT CHEM 8+ Cartridge (Abbott Laboratories, Chicago, IL, USA; $n = 5$ each group). The remaining blood was left at room temperature and allowed to clot before centrifugation at 2000 *g* for 15 min at which point serum was stored at −80°C. Urine was collected directly from the bladder via syringe and stored at −80°C. Gastrocnemius weights for male animals

alone were measured after stripping the muscle from the Achilles prior to tendon testing.

### *Ex vivo* muscle testing

The mechanical properties of diaphragm muscle strips were examined as in previous studies (Brashear et al., 2021; Smith & Barton, 2014) (experimental set-up, Fig. 4*B*). Diaphragm muscles with the surrounding rib cage were stored in Ringer's solution on ice with bubbling oxygen until mechanical testing was performed. Strips were cut from the diaphragm, and 7-0 sutures were tied onto the rib cage and at the myotendinous junction and the central tendon to connect it to a 300C-LR Dual-Mode motor arm and force transducer (Aurora Scientific). Optimum muscle length ($L_0$) was determined by performing twitches as the muscle was slowly stretched using the 701C high-power follow stimulator (Aurora Scientific), with the length set at the largest twitch amplitude (Moorwood et al., 2013). $L_0$ was measured using a calliper between the rib cage and the myotendinous junction at the central tendon. A passive mechanical testing protocol was performed where the muscles were cyclically strained 2.5% $L_0$ at 1 Hz for 5 s. This was repeated at 5, 7.5, 10 and 12.5% $L_0$. Immediately after 10% $L_0$ cyclic stretching the muscle was held at 10% strain for 2 min to obtain a stress relaxation curve prior to undergoing 12.5% $L_0$ cyclic strain. At the end of the 12.5% strain muscles were returned to $L_0$. A custom MATLAB script was used to fit a quadratic function to the peak stresses and to the nadir stress (following 2 min of relaxation). Dynamic stiffness was calculated as the tangent slope at 10% strain of a quadratic fit of the peaks and elastic stiffness as the tangent slope at 10% strain of the quadratic fit of the nadir, as previously described (Smith & Barton, 2014) (Fig. 4*B*).

After passive mechanics, an active fatigue protocol consisting of an isometric twitch contraction followed by 25 maximum isometric tetanic contractions 5 s apart, was performed on each diaphragm strip. After the 25th contraction there was a 5 min waiting period before a final isometric tetanic contraction was performed to assess muscle recovery. After completion of mechanical testing the muscle strip was isolated, and muscle mass was measured to determine physiological cross-sectional area (PCSA). The diaphragm was then flash-frozen in liquid nitrogen and stored at −80°C until the hydroxyproline assay was performed.

### Western blotting

Western blotting analysis of pyruvate dehydrogenase and oxidative phosphorylation proteins was performed on soleus muscle samples. Tissues were powdered on

dry ice, incubated on a shaker for 1 h in 200 μl of sucrose lysis buffer and centrifuged for 10 min at 8000 RPM and 4°C. The supernatant was collected, and a detergent-compatible protein assay (DC Protein Assay, Bio-Rad Laboratories, Hercules, CA, USA) was used to normalize protein concentrations across samples, which were diluted into Laemmli sample buffer and sonicated. Samples (12 μl) were loaded onto 4%–20% Criterion TGX Stain-free gels (BioRad) and were run at 200 V for 45 min. The gels were activated under UV light to fluorescently quantify total protein in each well. Proteins were transferred onto a nitrocellulose membrane at 100 V for 30 mins. Membranes were ponceau stained to confirm protein transfer, washed in Tris-buffered saline w/ 0.1% Tween (TBST) and blocked in 5% skim milk in TBST for 30 min. Membranes were then rinsed in TBST and incubated overnight at 4°C with the appropriate primary antibody diluted in TBST at 1:1000. The following day membranes were washed in TBST (3 × 5 min) and incubated with peroxidase-conjugated secondary antibodies in TBST at 1:10,000 for 1 h at room temperature. After secondary antibody incubation, Immobilon western chemiluminescent HRP substrate (Millipore, Hayward, CA, USA) was applied to the membranes for protein visualization using chemiluminescence. Images were taken using the ChemiDoc MP system, and bands were quantified through Image Lab 5.0 software (BioRad). The following primary antibodies were used: PDH-E1α (Santa Cruz Biotechnology, sc-377092); Total OxPhos Rodent WB Antibody Cocktail (abcam, ab110413). Full protein gel and blot images can be found in Appendix A2.

### Histology

Kidneys were sectioned at 10 μm thickness, stained with Masson's trichrome and imaged at 20× magnification by the UC Davis Centre for Genomic Pathology Lab. Images were reviewed by a clinical pathologist, and a subset of four slides from each $_{ADI}$CKD group were scored for interstitial fibrosis and tubular atrophy (Appendix A3 and A4).

### Collagen assays

Collagen content was measured using a hydroxyproline assay (Woessner, 1961). Kidney, muscle, Achilles and TA tendon samples were weighed pre- and post-drying at 120°C for 90 min (30 min for tendon) to determine water concentration. One centimetre lengths of cortical bone were cut from the middle of the femur, demineralized in 0.5 N HCl for 24 h, placed in deionized water, sonicated for 15 min, then rinsed five times in deionized water (Pang et al., 2021). Femurs were weighed once following demineralization and removal of marrow. All tissues were hydrolysed in 6 N HCl for 2 h, following which HCl was boiled off in a fume hood. Samples and hydroxyproline standards were then suspended in 200 μl of hydroxyproline buffer (173 mm citric acid, 140 mm acetic acid, 588 mm sodium acetate, 570 mm sodium hydroxide), and samples were diluted further in hydroxyproline buffer to keep collagen content within range of the assay. Next 150 μl of chloramine T solution was added to each sample, before being mixed and incubated for 20 min at room temperature. After this 150 μl of aldehyde-perchloric acid (60% 1-propanol, 5.8% perchloric acid and 1M 4-(dimethylamino) benzaldehyde) was added to each tube, and the samples were incubated at 60°C for 15 min. Samples were cooled for 5 min, and 200 μl of each was added to a clear-bottomed 96-well plate, and absorbance at 550 nm was measured using an Epoch Microplate Spectrophotometer (BioTek Instruments Limited, Winooski, VT, USA). Collagen content was then determined by comparing sample absorbance to the standard curve, assuming that hydroxyproline constitutes 13.7% of the amino acid composition of collagen. Nine TA tendon samples taken from female animals were too small to weigh accurately, leading to collagen concentrations above 100%; these samples were excluded from analysis.

### Bone micro-computed tomography

A subset of four femurs from each group were scanned using micro-CT (SCANCO Medical, μCT 35, Brüttisellen, Switzerland) with 10 μm nominal voxel size (X-ray tube potential = 55 kVp, current = 114 mA, integration time = 900 ms, number of projections = 1000/180°). Trabecular bone was analysed at the distal femoral metaphysis. Trabecular regions of interest (ROIs) were manually drawn on transverse images, excluding the cortical surface. The ROIs for the distal femoral metaphysis were located 1.5 mm from the start of the growth plate and extended 1 mm proximally. Analysis of cortical bone in the femoral diaphysis was performed by contouring transverse slices, with a 1 mm ROI (100 slices) cantered on the mid-point of the femur. Femur mid-point was calculated from length as derived from CT scan. Microstructural parameters were determined using the manufacturer's analysis software.

### Mechanical testing of femur mid-shaft

Three-point bending was used to determine structural and material properties of femora using a materials testing system (Instron, TA Instruments, New Castle, DE, USA; $n = 6$ for controls and $n = 7$ for $_{ADI}$CKD). The span length of the lower supports was 10 mm, and the femur was positioned to load the anterior aspect of the bone in tension. A 1–2 N preload was applied to ensure contact with the upper platen, then loading was applied at a

displacement rate of 0.01 mm/s until failure. The resulting force and displacement were recorded at 50 Hz and analysed to determine whole bone stiffness, yield force, ultimate force and post-yield displacement.

### Tendon mechanical testing and differential scanning calorimetry

Tendons were mechanically tested using a single-column tensile tester with a 100 N load cell, with a BioPuls Body Temperature Bath and BioPuls Submersible Pneumatic Side Action Grips (Model 68SC-1, Instron, Norwood, MA, USA). The left and right Achilles and TA tendons were thawed in PBS-soaked gauze at room temperature before testing. Achilles and TA tendons were isolated from one another by carefully separating the first tarsal (and TA) from the rest of the foot. Width and depth were measured at the narrowest point of each tendon using optical coherence tomography (OQ LabScope, Lumedica Inc, Durham, NC, USA), and cross-sectional area (CSA) was calculated (width $\times$ depth). All muscle was then removed from the proximal end of each tendon using forceps, and the top 4–5 mm was affixed inside a folded piece of 80-grit sandpaper (sand side in) using fast-drying cyanoacrylate. Tendons were then placed into the tensile tester with the sandpaper and either the entire foot (Achilles) or the first digit (TA) clamped into the top and bottom pneumatic grips (80 psi), respectively. Gauge length was measured using digital callipers. Samples were immersed in 0.9% PBS within the BioPuls bath held at 37°C, preconditioned using 10 cycles (0.10–0.25 N, 0.25 mm/s) and then elongated at 0.25 mm/s until failure (defined as an 80% decline in force). When not visually obvious, failure was confirmed by gently passing a pair of forceps through the remaining tendon tissue. Force and displacement were recorded throughout, from which the maximal tensile load (MTL; N) was determined as the highest tensile force obtained before failure – representing mechanical strength. Maximal apparent stiffness (N/mm) was calculated as the steepest slope (using a least-squares fit) of the load-displacement curve (excluding the preconditioning phase) – representing the greatest capacity of the tissue to resist elongation. Material properties, failure stress and maximal modulus were calculated by normalizing MTL to CSA and maximal stiffness to CSA and initial length, respectively. Individual and group-averaged stress/strain traces are presented in Appendix A5. After tensile testing a 4 mm biopsy punch was used to cut a similarly sized piece from each tendon. Each tendon was then lightly blotted to remove excess liquid, placed on the bottom of a Tzero aluminium pan and sealed using an aluminium hermetic lid and Tzero press. Each sample was then weighed and placed into a differential scanning calorimeter with an autosampler (Discovery, TA Instruments, New Castle, DE, USA). After equilibration at 40°C samples were heated from 40 to 90°C at a rate of 10°C/min. The temperature at which material transitions began to occur (onset temperature) was determined as the temperature where the tangent of the melt curve intersected baseline. The temperature at which 50% of the material transitions have occurred (peak temperature) was determined as the point of peak heat flow, and the amount of energy needed to transition all of the material (enthalpy) was calculated as the area under the melt curve.

### Statistics

Muscle and tendon data were screened for outliers using the ROUT method and a Q = 1%. Outliers and any other values dependent on outlying measurements were removed from analysis. Bone and serum data were not screened for outliers due to the low *n*. Evaluation of the effects of adenine diet, sex or interaction between the two was evaluated using two-way ANOVA, with the exception of *in vivo* performance testing for which analysis of covariance (ANCOVA) was used to control for baseline performance. If an interaction was detected, a *post hoc* Fisher's least significant difference (LSD) test was used to evaluate the effect of adenine diet in males and females separately. Unpaired *t* tests were used to compare protein levels (determined by western blot) across diet interventions for each sex separately to ensure that comparisons were made only within a single blot. All statistics were performed using either GraphPad Prism (version 10.4.2, GraphPad Software, Boston, MA, USA) or MedCalc statistical software (version 23.2.8, MedCalc Software Ltd, Ostend, Belgium).

## Results

### Nine weeks of adenine feeding induced renal dysfunction in growing rats

All animals, control and adenine fed, survived the entire course of the study. Over 9 weeks, the increase in body mass in adenine-fed female rats mirrored that seen in female controls. However male controls gained significantly more weight than the male adenine group (585 *vs.* 439 g at week 8, $P < 0.001$, Fig. 1). This difference in weight is likely due to a decreased consumption of food in the adenine-fed males compared to control males, leading to a similar but slightly lower estimated adenine dosage compared to the adenine-fed females (Fig. 1). Masson's trichrome staining of kidneys showed an increase in collagen deposition, 2,8-dihydroxyadenine crystals, calcium phosphate crystals, neutrophil infiltration, tubule dilatation and proximal tubule hyper-

trophy in the adenine-diet fed animals (Fig. 2*A* and Appendix A4). Adenine feeding also led to a significant increase in kidney mass that coincided with a greater collagen concentration (by hydroxyproline assay) and water content (Fig. 2*B*). This increase in kidney mass was more pronounced in the left kidney (∼117%) compared to the right (∼57%). Serum creatinine and blood urea nitrogen (BUN) were elevated in adenine-fed animals compared to controls (2.09 *vs.* 0.68 and 98.1 *vs.* 28.4 mg/dl, respectively), an effect that was more pronounced in male animals, with several exceeding the upper limit of detection for BUN (140 mg/dl). Haematocrit was significantly decreased in the adenine-fed group (33.7% *vs.* 47.3%, Fig. 2*C*). Notably, blood glucose was elevated in male controls alone, coinciding with their higher food consumption and continuous weight gain. Collectively these results demonstrate that 9 weeks of adenine feeding led to significant renal dysfunction, which tended to be more severe in male animals. The histological and humoral phenotype of adenine-fed animals recapitulates characteristics of human CKD and is consistent with previous descriptions of this model (Diwan et al., 2013, 2014; Yokozawa et al., 1986), successfully establishing the presence of renal dysfunction.

### ADICKD causes bone mineral wasting and loss of mechanical strength

Femurs from all animals were imaged using micro-CT to evaluate the effects of ADICKD on bone structure and mineral content. Animals with ADICKD had significantly lower cortical thickness and tissue mineral density compared to healthy controls. At the same time collagen content (via hydroxyproline assay) was unchanged (Fig. 3*A*). Femur cross-sections from ADICKD animals were visibly porous and had significantly greater pore area and average pore size. However this was more variable in female animals, with several appearing to be unaffected (Fig. 3*A*). There was no effect of ADICKD on trabecular bone volume fraction (BV/TV), but the disease resulted in an increase in trabecular thickness (Fig. 3*B*). Trabecular bone tissue mineral density; however, was lower in ADICKD animals, suggesting poorer bone quality (Fig. 3*B*). Mechanical testing showed that ADICKD rats did not differ from healthy controls in terms of stiffness or post-yield displacement. However yield force and ultimate force were significantly lower in male ADICKD animals (Fig. 3*C*).

### ADICKD decreased muscle strength but not endurance

Muscle strength and endurance were evaluated both *in vivo* (plantar flexors) and *ex vivo* (diaphragm). There was a significant effect of time on plantar flexor strength, with all groups increasing in maximal torque from pre-to-post diet intervention (50% ± 35%, *P* < 0.001). Strength was significantly lower in ADICKD after correcting for baseline differences (Fig. 4*A*). There was no significant effect of time on plantar flexor endurance, determined as the average torque output over 180 contractions, but there was a sex-by-time interaction driven by greater baseline endurance in female animals that declined over time (Fig. 4*A*). There were no effects of diet on plantar flexor endurance, or soleus weight, collagen concentration or water concentration (Fig. 4*A*). Gastrocnemius weight was significantly lower in male ADICKD compared to controls (1.7 ± 0.2 *vs.* 2.3 ± 0.3 g, *P* < 0.001) but was not measured in female animals. Muscle electron transport complex levels were not affected by ADICKD; however, PDH levels were lower in diseased males (Fig. 4*A*). A strip of diaphragm muscle was excised and mechanically tested *ex vivo* for both active (stimulated) and passive (relaxed) tension generation. There was no effect of diet on diaphragm peak force, fatigue or recovery (Fig. 4*B*). Passive mechanical analyses, which were evaluated using a stress-relaxation test (passive mechanics, Fig. 4*B*), demonstrated a significant increase in both dynamic and elastic stiffness in the diaphragm muscle of ADICKD animals (Fig. 4*B*). Although there is no significant interaction, this appears to be driven by lower control values in the males.

### ADICKD decreased tendon failure stress without changing cross-sectional area or collagen content

Both the Achilles and TA tendons were excised and tested for strength and function, collagen concentration and extracellular matrix stability. Because these two tendons have different functions – the Achilles is an energy-storing tendon and the TA is a positional tendon – their mechanical and material properties, as well as their response to load and disease, may differ. Therefore results were analysed separately. There was no effect of diet on Achilles tendon mechanics, collagen concentration or measures of extracellular matrix stability (onset temperature, peak temperature and enthalpy; Fig. 5*A*). There was, however, an interaction effect on failure stress and CSA. *Post hoc* testing revealed that in females Achilles failure stress was significantly decreased in ADICKD (−25%) and CSA was significantly increased (+26%, Fig. 5*A*). In the TA tendon there was a significant effect of diet on failure stress, with ADICKD tendons being 24% weaker. There were not significant decreases in MTL (−15%, *P* = 0.0998) or stiffness (−17%, *P* = 0.0632) in the ADICKD group (Fig. 5*B*). No significant effect of diet was found on TA tendon CSA, collagen concentration or measures of extracellular matrix stability.

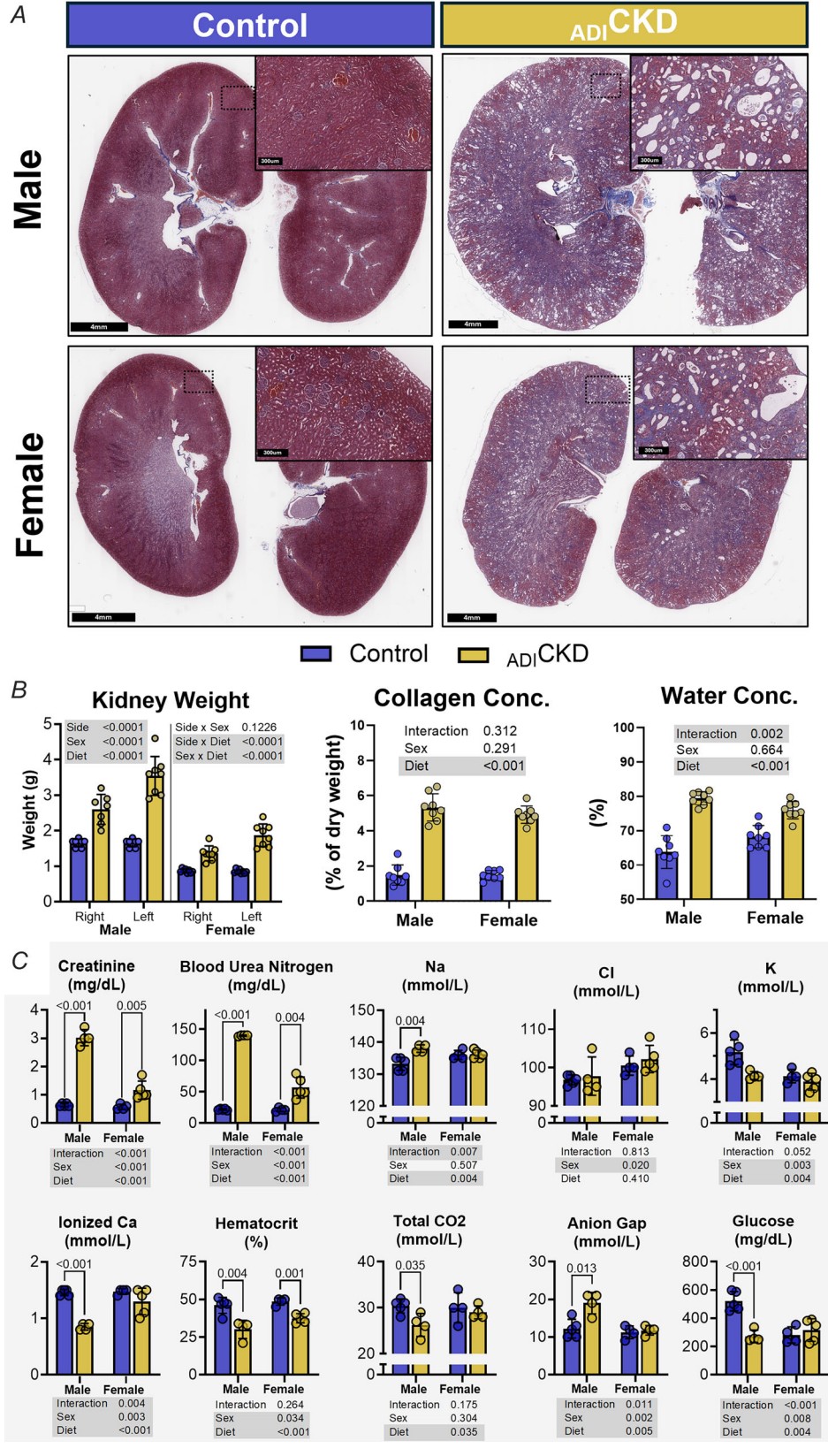

**Figure 2. Adenine feeding for 9 weeks leads to CKD phenotype in rats that is more severe in males**
*A*, Masson's trichrome staining of kidneys from control and ADICKD rats, with collagen deposition shown in blue.
*B*, kidney weight and collagen and water concentrations from hydroxyproline assays (*n* = 8/group/sex). *C*, Blood

markers measured at collection (week 9) using an i-STAT with CHEM 8+ cartridges ($n = 5$/group/sex, $n = 4$ $_{ADI}$CKD males due to a cartridge error). Statistics were calculated using two- or three-way ANOVA, with significant effects highlighted in grey. *Post hoc* Fisher's least significant difference (LSD) tests were performed when significant interactions were observed. $_{ADI}$CKD, adenine-diet-induced chronic kidney disease; Na, sodium; Cl, chloride; Ca, calcium.

## Discussion

The goal of this study was to evaluate the use of $_{ADI}$CKD in rats for studying CKD-related tendon dysfunction. Nine weeks of adenine feeding successfully recapitulated the CKD phenotype previously reported in the literature (Diwan et al., 2013; Yokozawa et al., 1986). This includes significant kidney dysfunction evidenced by kidney crystalline deposits, neutrophil infiltration and interstitial fibrosis, as well as elevated blood creatinine, BUN and lower haematocrit (Fig. 2). Consistent with our hypothesis, rats with $_{ADI}$CKD had significantly weaker tendons compared to controls (∼25% lower failure stress, Fig. 5). This detrimental effect on tendons is evident in $_{ADI}$CKD rats in the absence of pharmaceutical treatment, suggesting that kidney impairment itself is detrimental to tendon health. To our knowledge this is the first direct evidence of altered tendon mechanics in a model of CKD.

Muscle wasting (Chatzipetrou et al., 2022), osteoporosis (Pazianas & Miller, 2021) and poor exercise capacity (Roshanravan et al., 2013) are consistently seen in CKD, and such musculoskeletal dysfunction and consequent physical inactivity are highly predictive of mortality in CKD populations (Painter & Roshanravan, 2013; Roshanravan et al., 2013, 2017). This is the first study known to the authors to combine both functional and structural analyses of muscle, and bone in a CKD model into one paper. As muscle–bone crosstalk in CKD has recently become a topic of interest (Leal et al., 2021; Wong & McMahon, 2023) the data presented here support the use of this model to mechanistically explore this area. Furthermore the analysis of tendon tissue is completely novel and when integrated with the muscle and bone data creates a platform for comprehensive evaluation of the effects of the CKD on musculoskeletal function.

Officially termed 'renal osteodystrophy', bone dysfunction in CKD falls under the umbrella of CKD-mineral and bone disorder (CKD-MBD), which differs from bone dysfunction seen in other diseases (Moe et al., 2006; Pazianas & Miller, 2021). General characteristics of CKD-MBD include high circulating FGF-23 and parathyroid hormone, low concentrations of active vitamin D and altered calcium and phosphate homoeostasis (Pazianas & Miller, 2021). This leads to decreased volume, thinning and increased porosity in cortical bone, with less-severe and sometimes opposite effects seen in trabecular bone (Malluche et al., 2011, 2018; Nickolas et al., 2013), collectively resulting in an increased risk of fracture (Alem et al., 2000). In line with previous studies in rats (Saito et al., 2021) and mice (Metzger et al.,

2021), we found that $_{ADI}$CKD led to significant bone pathology, including decreased bone mineral density and volume in cortical bone and increased porosity (Fig. 3*A*). Further recapitulating human disease (Malluche et al., 2011, 2018; Nickolas et al., 2013), we observed an increase in thickness in trabecular bone despite a concomitant decrease in mineral density (Fig. 3*B*). In accordance with the mouse work from Metzger and colleagues (Metzger et al., 2021), the bone phenotype we observed was slightly more severe in males than females. This coincided with decreased mechanical properties in the $_{ADI}$CKD males (Fig. 3*C*), which is evident in human CKD in the form of higher rates of fracture (Alem et al., 2000). We also observed a decrease in serum ionized calcium (Fig. 2*C*), and although we did not measure parathyroid hormone or FGF-23, others have found these to be elevated in $_{ADI}$CKD rats (Neven et al., 2015; Saito et al., 2021). Together our results suggest that 9 weeks of adenine feeding in rats effectively recreates relevant aspects of CKD-MBD in humans.

Cachexia, or muscle–protein energy wasting as a result of disease, is a well-established comorbidity in CKD, with a large number of confounders that make it difficult to study in humans (Carrero et al., 2013; Hanna et al., 2020; Wang et al., 2022). Several studies using animal models of CKD have reported decreases in muscle mass (Fusagawa et al., 2023; Kim et al., 2021; Momb et al., 2022; Saito et al., 2021), *in vivo* strength and endurance (Fusagawa et al., 2023), and *ex vivo* single-fibre force (Momb et al., 2022, 2023). These negative effects of CKD on muscle vary by sex and muscle (Kim et al., 2021) and are undoubtedly dependent on the testing protocol and model used. To our knowledge this is the first study to evaluate muscle function *in vivo* and *ex vivo* in the rat $_{ADI}$CKD model. We found that plantar flexor strength decreased by 13% in $_{ADI}$CKD with no concomitant change in soleus mass (Fig. 4*A*). The gastrocnemius, which is the larger plantar flexor in rats, was 26% smaller in the $_{ADI}$CKD males, suggesting that muscle atrophy may be responsible for the decline in strength; however this measure was added midway through collections and therefore not confirmed in females. We designed our *in vivo* muscle function testing to be similar to that of Fusagawa and colleagues, who showed that 8 weeks after inducing CKD using a 5/6 nephrectomy, male rats had significant reductions in plantar strength (due to muscle atrophy) and also plantar flexor endurance (Fusagawa et al., 2023). Our study did not find any sign of reduced muscle endurance using the same protocol. We did find that $_{ADI}$CKD led to decreased levels of PDH in the soleus of male animals,

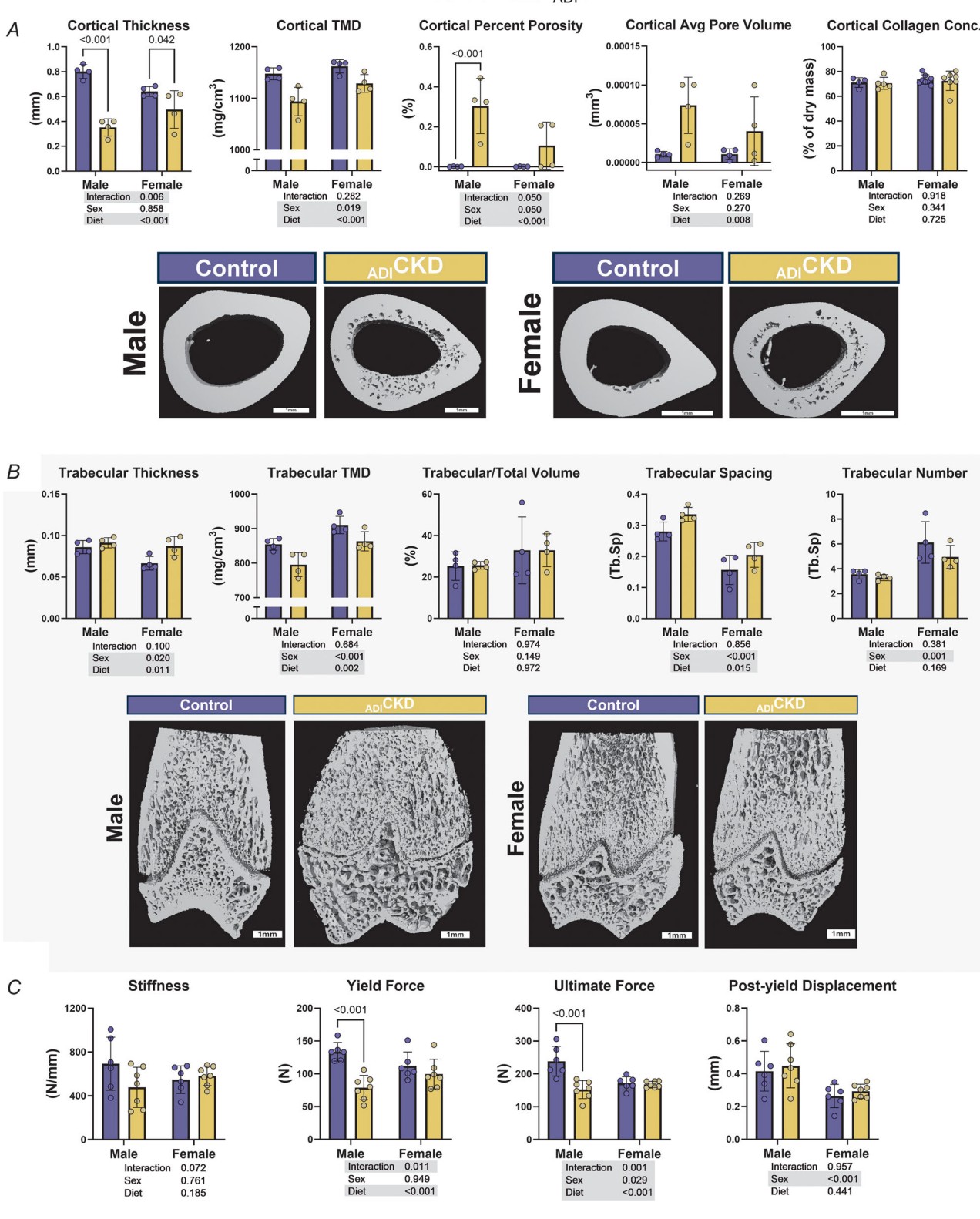

**Figure 3. Bone morphology and mechanical strength are altered in ADICKD**

*A*, cortical bone micro-computed tomography (CT) measurement and images from femurs of control and ADICKD rats (*n* = 4/group/sex). *B*, trabecular bone micro-CT measurement and images from femurs of control and ADICKD rats (*n* = 4/group/sex). *C*, three-point-bending mechanical analysis of femurs (*n* = 6/controls, *n* = 7/ADICKD). Statistics were calculated using two-way ANOVA, with significant effects highlighted in grey. *Post*

*hoc* Fisher's least significant difference (LSD) tests were performed when significant interactions existed. ADICKD, adenine-diet-induced chronic kidney disease; TMD, tissue mineral density; Conc., concentration.

but no effect was observed in female rats or in electron transport chain complexes (Fig. 4*A*). Animal (Tamaki et al., 2014; Thome et al., 2019, 2021) and human (Xu et al., 2020) studies that have reported decreased PDH activity in CKD, which has been shown to contribute to mitochondrial dysfunction and impaired endurance performance; however, we measured only PDH content and did not observe endurance deficits in the current work.

We also completed an *ex vivo* assessment of diaphragm muscle active and passive tension to evaluate muscle function independent of circulation and innervation. Diaphragm function may be less influenced by any potential disease-related changes in activity levels and is particularly relevant as CKD is associated with a number of respiratory comorbidities (Bollenbecker et al., 2022). Although we did not find significant effects of ADICKD on diaphragm active mechanics, there were differences in dynamic and elastic stiffness (i.e. passive mechanics, Fig. 4*B*). Passive mechanics measures the resistance of muscle to lengthening in a relaxed state and can be indicative of changes in muscle, such as fibrosis or altered matrix orientation and cross-linking (Wohlgemuth et al., 2023). The effect on passive mechanics was driven by the apparent deficits in control males, where signs of obesity and metabolic disorder were evident. Our study found no increase in collagen content in ADICKD soleus or diaphragm, suggesting that changes in passive mechanics were not due to fibrosis (Fig. 4*A*,*B*). Pathological fibrosis has been reported in the muscle of CKD patients in two studies by one group (Abramowitz et al., 2018; Brightwell et al., 2021), and there is only non-quantitative evidence of muscle fibrosis in animal CKD models (Alcalde-Estévez et al., 2021; Dong et al., 2017). This is potentially counter-acted by the fact that obesity, which is common in CKD, can decrease collagen production in muscle, bone and tendon (Lin et al., 2024). Collectively our results suggest that 9 weeks of adenine feeding in 8-week-old Sprague–Dawley rats induces muscle pathology in some, but not all, muscles, as evidenced by declines in strength and, to a limited extent, mass and passive mechanics. These results do not align with a recent study by Lair and colleagues, which suggested that common rodent CKD models do not induce cachexia (Lair et al., 2024). However the Lair et al. study used a 3 week adenine diet model that is significantly shorter than most protocols and failed to induce severe uremic build-up (creatinine < 0.51 mg/dl), which is at odds with the literature (Fusagawa et al., 2023; Kim et al., 2021; Momb et al., 2022; Saito et al., 2021).

Although CKD-related muscle and bone disorders have been subject to numerous investigations, almost no data exist regarding tendon pathology in CKD or its mechanisms. In this study we found that rats with ADICKD had significantly weaker tendons (lower failure stress) than controls (Fig. 5), supporting use as a model for CKD-induced tendon pathology. Failure stress is a material property indicative of tissue quality. The decrease in failure stress found in this study, therefore, suggests that ADICKD tendons have reduced quality and are not simply smaller. As opposed to other organs that are primarily composed of parenchymal cells (liver, kidney, heart, muscle, etc.) tendons are composed of less than 20% cellular mass. The extracellular matrix, which is produced and secreted by tendon cells, makes up the remaining 80% of tendon mass and is central to tendon function (force transfer). The majority of the extracellular matrix in tendons comprises collagen, predominantly collagen I, which aligns along the longitudinal axis and provides tensile strength (Amiel et al., 1983). Despite the decrease in material properties there were no differences in tendon collagen concentration between control and ADICKD animals. Additionally there was no effect of ADICKD on matrix thermal stability measures (Fig. 5), which can be used to roughly gauge collagen microstructure (cross-linking, fibril packing density, etc.) (Miles et al., 2005; Willett et al., 2010). Thus the exact structural change leading to the decrease in tendon material properties remains unknown.

Mechanical properties (MTL and stiffness) are more functionally important characteristics as they define the absolute ability of the tendon to resist elongation. An important aspect of this study is that it was conducted in a strain of rats that continue to increase body weight throughout life (Fig. 1) and commonly develop insulin resistance (Cummings et al., 2008). Tendon mechanical properties increase across development (Steffen et al., 2023), likely to meet the functional demands of concurrent increases in body weight. Thus over the course of this study TA and Achilles tendons should have grown in length to match any longitudinal bone growth and in CSA to maintain or increase tendon mechanical properties. In TA tendons there were no significant decreases in mechanical properties in ADICKD. Sub-jectively ADICKD tendons did not appear to fully adapt to maintain mechanical properties in response to body growth or to compensate for their decreased material properties (Fig. 5*B*), suggesting that tendon homoeostasis was not maintained. However, a study with greater power would be necessary to validate this hypothesis. Conversely in the Achilles there was a significant increase in CSA in female rats with ADICKD, resulting in no apparent deficit in MTL or stiffness. Because the Achilles tendon

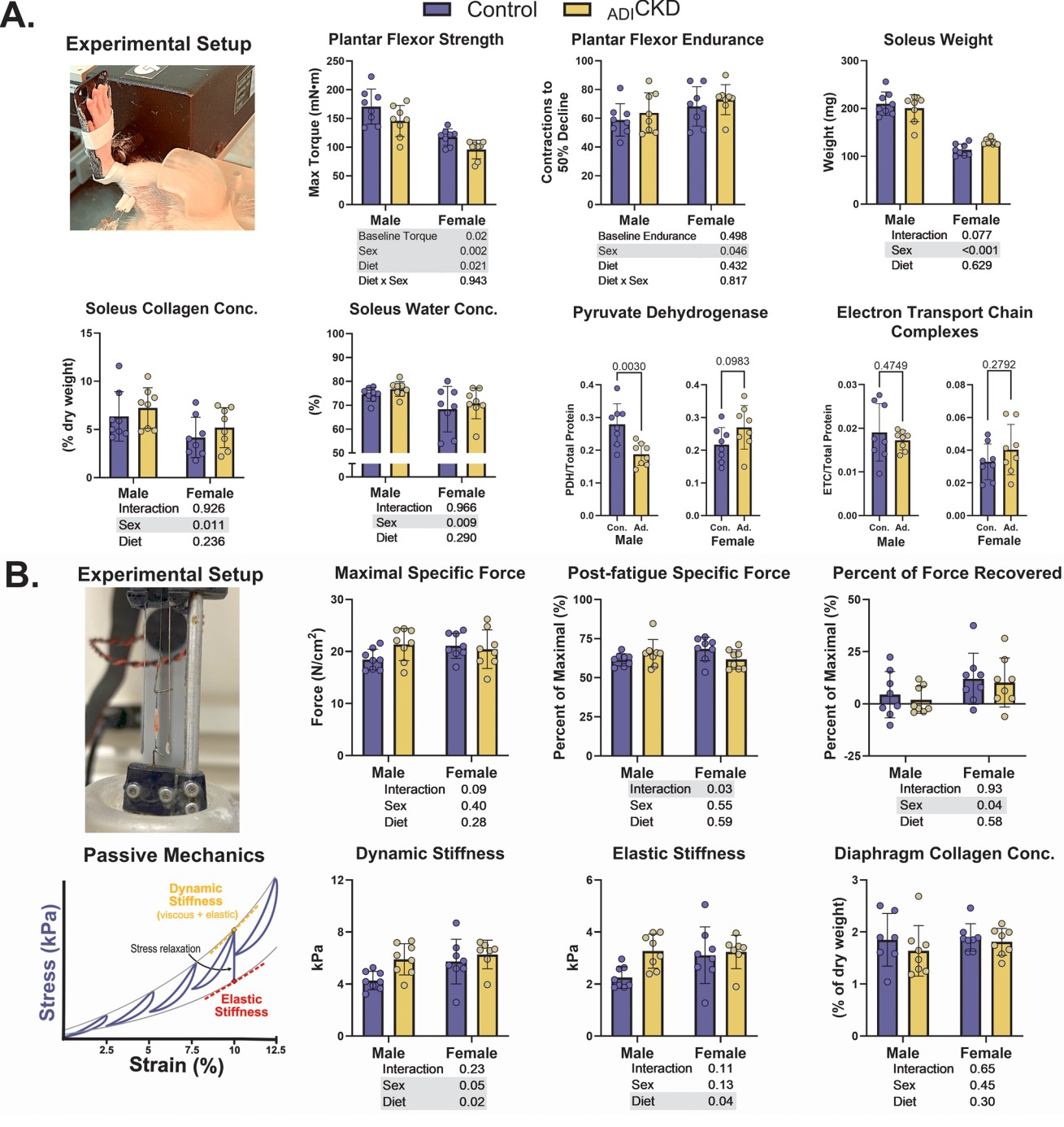

**Figure 4. Plantar flexors are weaker but not more fatigable, and diaphragm is stiffer but not more fibrotic in ₐDICKD**

*A*, *in vivo* assessment of plantar flexor strength and endurance was measured via muscle electrical stimulation to elicit maximal torque and over 180 repeated submaximal contractions, as well as soleus mass, collagen and water content from a hydroxyproline assay, and pyruvate dehydrogenase and electron transport chain protein levels from western blotting (*n* = 8/group/sex). *B*, *ex vivo* assessment of a strip of diaphragm muscle during active and passive mechanical protocols, along with collagen content via hydroxyproline assay (*n* = 8/group/sex). Statistics were calculated using two-way ANCOVA (strength and endurance) or ANOVA, with significant effects highlighted in grey. *Post hoc* Fisher's least significant difference (LSD) tests were performed when significant interactions existed. ₐDICKD, adenine-diet-induced chronic kidney disease; Conc., concentration.

is a load-bearing tendon, whereas the TA tendon is a positional tendon, during ambulation the Achilles tendon will experience loads close to body weight, and the TA will only be exposed to the load of the foot. Therefore load-dependent signalling processes may have played a role in maintaining mechanical properties in the Achilles,

but this requires further confirmation. Of note there was not a significant effect of $_{ADI}$CKD on failure stress in the male Achilles tendons (Fig. 5*A*). This is likely due to the larger body size of the control males (585 *vs*. 439 g for $_{ADI}$CKD) and decrease in insulin sensitivity (increased blood glucose; Fig. 2*C*). Because insulin resistance is well

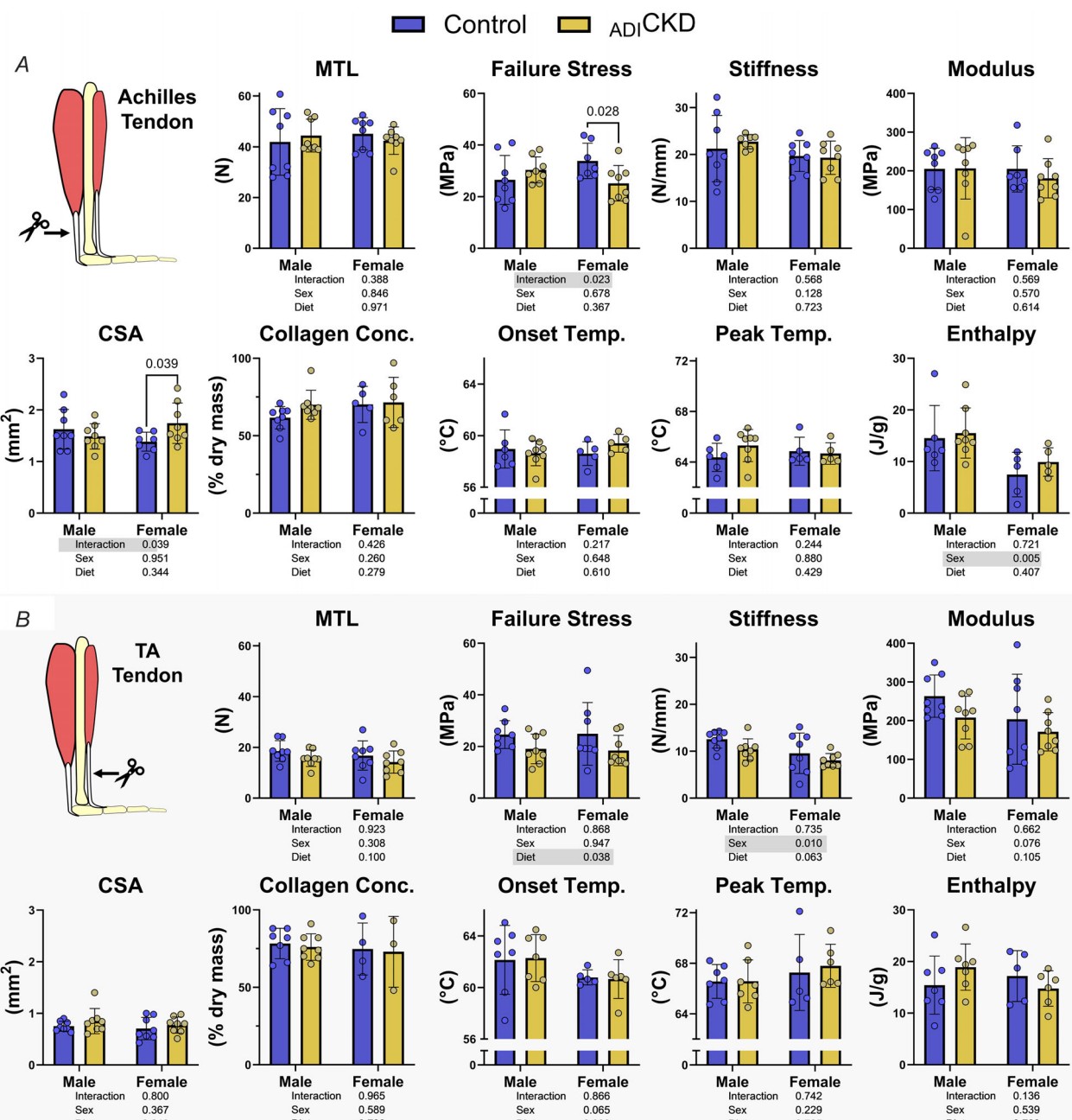

**Figure 5. Tendons are significantly weakened in rats with $_{ADI}$CKD**

*Ex vivo* assessment of tendon mechanical and material properties under tensile strain to failure, collagen content via hydroxyproline assay and matrix thermal stability via differential scanning calorimetry in (*A*) Achilles and (*B*) tibialis anterior (TA) tendons from control and $_{ADI}$CKD rats (*n* = 8/group/sex). Statistics were calculated using two-way ANOVA, with significant effects highlighted in grey. *Post hoc* Fisher's least significant difference (LSD) tests were performed when significant interactions existed. $_{ADI}$CKD, adenine-diet-induced chronic kidney disease; MTL, maximum tensile load; CSA, cross-sectional area; Conc., concentration, Temp., temperature.

known to impair tendon function (Galeski et al., 1977; Li et al., 2013), interpretation of the Achilles tendon data from the male rats is difficult.

The only large-scale analysis to date of tendon rupture in CKD (Humbyrd et al., 2018) suggests that rupture is less common in men than in women (adjusted incidence rate ratio, aIRR = 0.75). Although our study shows less-consistent effects of CKD on tendon in male rats (in agreement with Humbyrd), we believe this is due to the negative effects of obesity and inactivity in the male control group rather than a less-pronounced effect of CKD. Consistent with this hypothesis, Humbyrd and colleagues also found that rupture risk increased with increasing body mass index (BMI) (reference BMI: 0–24.99, BMI = 25–29.99: aIRR = 1.25, BMI = 30–39.99: aIRR = 1.56, BMI ≥ 40: aIRR = 1.81). Importantly the Humbyrd study included only ESRD patients who were on dialysis or had already received a kidney transplant, which is different from our model.

In the literature, hypotheses as to why CKD leads to tendon dysfunction include chronic acidosis (Finlayson et al., 1964), increased parathyroid hormone (de Franco et al., 1994; Jones et al., 1996) and beta-2 microglobulin amyloidosis (Portales-Castillo et al., 2020). The most popular explanation appears to be elevated parathyroid hormone, which has been shown by two studies to be higher in dialysis patients with tendon rupture than those without (Jones et al., 1996; Pichone et al., 2024). However, the parathyroid hormone level of the rupture group in the first study (Jones et al., 1996) (1802 pg/ml) was roughly the same as the non-ruptured group in the second study (Pichone et al., 2024) (1728 pg/ml), bringing into question validity of this finding. This highlights the pitfalls of observational studies and speaks to the need for well-controlled experimental research on the topic. Regardless of the cause, spontaneous tendon ruptures are debilitating injuries that occur as a result of pre-existing tendon pathology. In fact in a study of biopsies from 891 spontaneously ruptured tendons not a single healthy structure was found, whereas only one third of 445 tendons from healthy individuals had pathological changes (Kannus et al., 1991). Therefore significant tendon pathology likely precedes spontaneous rupture seen in many CKD patients and may contribute to the high prevalence of reported musculoskeletal pain in CKD (Lambourg et al., 2021) that reduces physical activity and quality of life. Unfortunately, tendons are notoriously hard to investigate *in vivo*, which has led to equivocal results from a number of low-quality studies on the impact of CKD on tendon size and stiffness (using ultrasonography) (Caglar et al., 2019; Hekimoglu et al., 2019; Kural Rahatli et al., 2021; Mutlu et al., 2021; Tang et al., 2022; Teber et al., 2015). To date, most scientific literature on the topic of tendon dysfunction in CKD consists of case studies of one or a few individuals, which limits inference of potential mechanisms. Due to the scarcity of non-invasive techniques for interrogating tendon function and the necessity of biological samples for mechanistic research, an animal model is necessary to progress towards treatment. Rats are an effective model system as they are relatively low cost, easy to handle, have high homology with humans and are frequently used for the study of tendinopathy (Warden, 2007). Rats are also preferred over mice due to their larger tendon size, trainability and temperament (Warden, 2007). The use of rat models allows excision of whole tendons at sacrifice, enabling accurate and simple mechanical testing. The tested tendon samples can then be evaluated biochemically, allowing mechanistic and functional inter-rogation of the same tendon.

### Additional considerations

As with all research there were limitations in this study. First, CKD typically affects adults, but here we used young rats (8 weeks) that had not yet fully developed. However, because tendon development involves similar mechanisms to repair (Nourissat et al., 2015), this approach may offer useful insight into maladaptive responses that lead to degeneration in adult CKD. Still, conducting a similar study with skeletally mature animals may be more congruent with human disease. Second, in humans kidney disease must last at least 3 months to be considered chronic. Our intervention lasted just over 2 months, and we did not determine when kidney dysfunction onset occurred. Thus, whether we were testing acute or chronic kidney dysfunction is unclear. Despite this limitation, the intervention was sufficient in length to induce significant tendon dysfunction, which highlights the interconnection between the two pathologies. Third, the male controls in this study ate significantly more than the males on the adenine diet, leading to large differences in body weight and blood glucose. Though not quantified, these animals likely had much greater fat accumulation as well and thus may have developed other comorbidities impairing their use as 'healthy' controls. Any follow-up studies may need to pair-feed animals to maintain similar body weights or rely on the use of the female animals, which maintain stable weights through adulthood and have been historically understudied.

As for the interesting finding that in $_{ADI}$CKD the left kidney is consistently significantly larger than the right, a potential explanation is that this occurs due to differences in renal blood flow. The left kidney has a shorter artery and longer vein, meaning there is less resistance to afferent flow and more resistance to efferent flow, whereas the right kidney has the opposite. This could potentially lead to differences in glomerular pressure and therefore hyper-

trophy, as our histology indicates obstruction induced dilatation. However, this is highly speculative and at the moment the authors have no definitive evidence for this.

Among our three musculoskeletal tissues, it may be noted that bone is the only one where the greater burden of CKD in males appeared to result in more severe pathology. The lack of apparent dose response in muscle and tendon to CKD might suggest that the pathobiology of kidney disease occurs at lower levels of disease, or that the metabolic impact of hyperphagia and the higher BMI was more detrimental to muscle and tendon than bone. Further, the mechanisms that lead to bone dysfunction in CKD likely differ from those driving the loss of muscle and tendon function. In fact, a surprising finding by Humbyrd and colleagues was that among individuals with ESRD, those with osteoporosis were far less likely to experience spontaneous Achilles rupture than those without (aIRR = 0.14) (Humbyrd et al., 2018). Spontaneous tendon ruptures in CKD have been reported in many different tendons of the upper and lower extremities, often occurring in load-bearing tendons of the knee and ankle – although there is at least one report of a torn medial obliquus abdominus internus (Basic-Jukic et al., 2009). Our finding that the TA tendon was more consistently affected by CKD than the load-bearing Achilles should be evaluated in context with two caveats. One, tendon loading within healthy bounds results in positive tissue adaptation that may counteract the negative effect of disease (e.g. the benefits of exercise) (Chen et al., 2020). Two, although adult tendons are relatively stable (i.e. low protein turnover) and may need external stimulus to initiate pathology, these animals developed kidney disease while their musculoskeletal system was still growing and maturing. Thus, in this model the negative effects of CKD on tendons may be evident in more locations as CKD may impair normal tendon development. Nevertheless, the finding of worse material properties in TA tendon is good evidence for a direct systemic effect of CKD on tendon tissue, as opposed to confounding from other CKD symptoms such as sedentarism.

## Conclusion

Tibialis anterior tendons from rats with $_{ADI}$CKD displayed evidence of disease-induced dysfunction, including significantly lower failure stress compared to controls. Additionally, inducing $_{ADI}$CKD in young Sprague–Dawley rats led to changes in kidney structure and function and bone pathology that mimics human CKD. Given the interdependency of tendon and bone function, the concurrence of CKD-MBD symptoms amplifies the relevance of the $_{ADI}$CKD model for use in CKD-related tendon dysfunction research. Significant

muscle impairment was also observed in this study, supporting the model's future use for comprehensive studies of CKD-related musculoskeletal dysfunction. This initial investigation did not detect any changes in tendon structure, such as altered collagen content or matrix stability, necessitating more mechanistic follow-up. Importantly, although several drugs commonly used to treat CKD patients, including corticosteroids, fluoroquinolone antibiotics, and angiotensin II receptor agonists, have been reported to increase tendon rupture risk (Nyyssönen et al., 2018), we believe the current data provide the first evidence of a direct impact of CKD on tendon function – laying the groundwork for further research.

## Appendix

Figures A1–A5

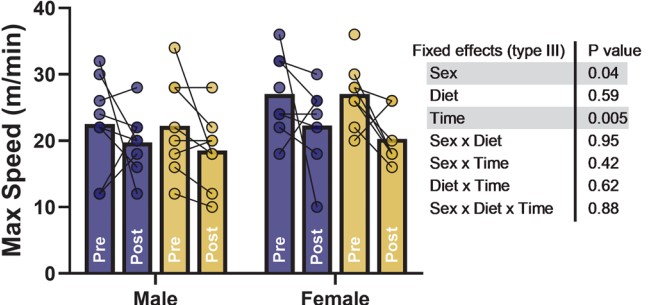

**Figure A1. Exercise tolerance test**

Exercise tolerance testing was completed prior to (pre) diet initiation and following 8 weeks on adenine or control diets (post). Baseline test performance was used to separate rats into groups with equivalent maximal running speeds. Both pre and post tests were completed with the same protocol as follows. Rats were accustomed to the treadmill by running for 5 min per day on a 10-degree grade for four consecutive days at 10, 15, 20 and 25 m/min, respectively. Each time the rats were run they were allowed to acclimate to the treadmill space for 3 min. During this time the electrical grid was on to discourage them from residing there, but the treadmill was not moving. To encourage running rats were prodded with a soft-bristled brush when they fell back onto the back quarter of the treadmill. On the fifth day the first two stages of the exercise tolerance were performed with the electrical grid on (1 Hz and 2 mA stimulation). On the sixth day the entire exercise tolerance test was performed, consisting of running on a 10-degree grade beginning at 10 m/min and increasing by 2 m/min every 3 min until volitional fatigue (sitting on the electrical grid > 5 s for females and > 10 s for males) or until a running gait could no longer be maintained (e.g. continual hopping on and off the grid). At cessation the electrical grid was immediately switched off, the animal was removed from the treadmill and the maximal speed achieved, as well as the total running time, were recorded. Animals did not show signs of exhaustion, and their righting response was not slowed. There was an effect of time and sex on maximal running speed; however there was no effect of disease status.

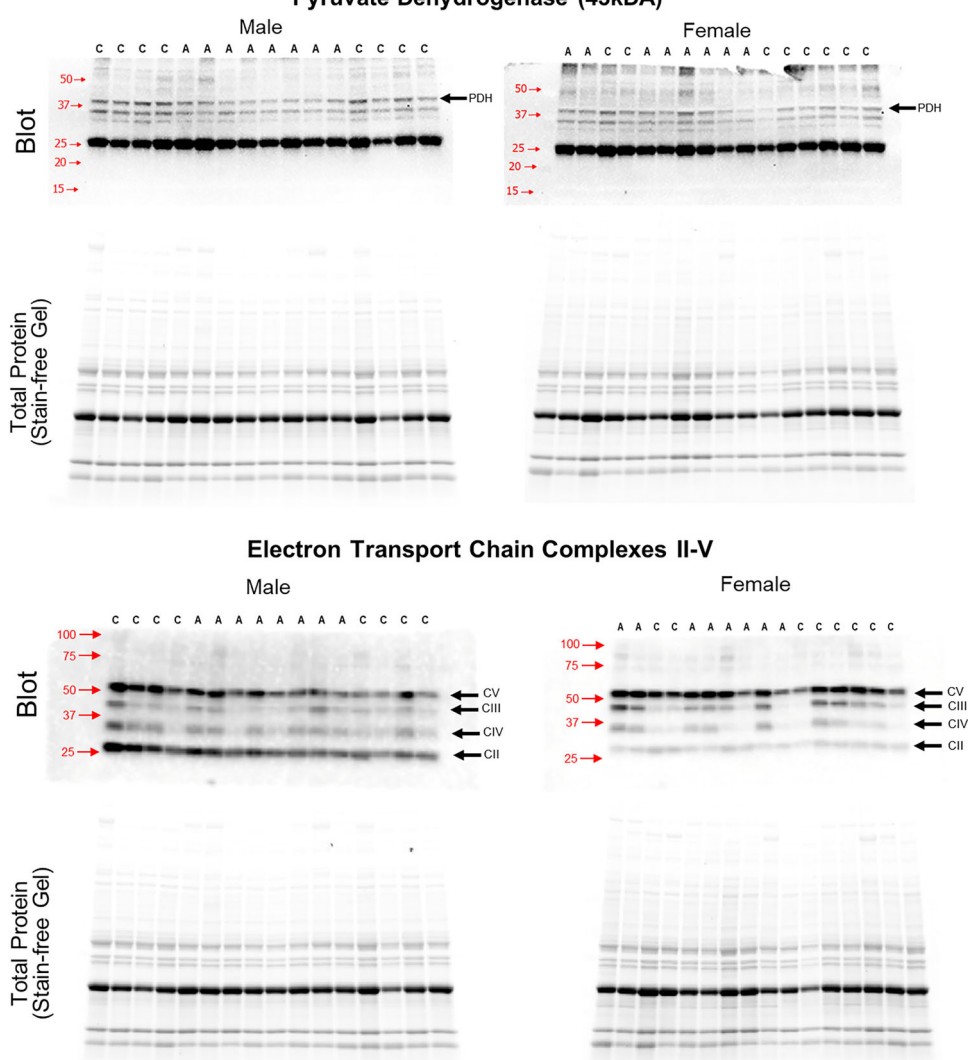

**Figure A2. Western blots**

Western blotting full membranes and gels from soleus muscles of control (C) and adenine-diet-induced chronic kidney disease (ADICKD) (A) rats.

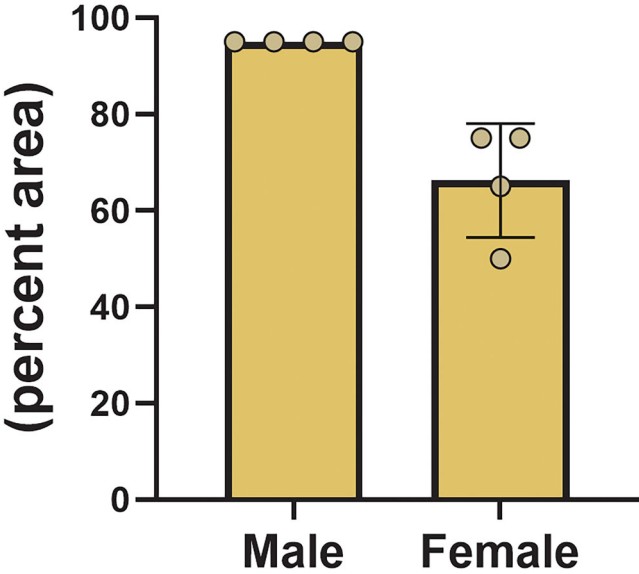

**Figure A3. Interstitial fibrosis and tubular atrophy scoring**
Scoring of adenine-diet-induced chronic kidney disease ($_{ADI}$CKD) kidneys from four rats from each disease group completed by a clinical pathologist (JYJ).

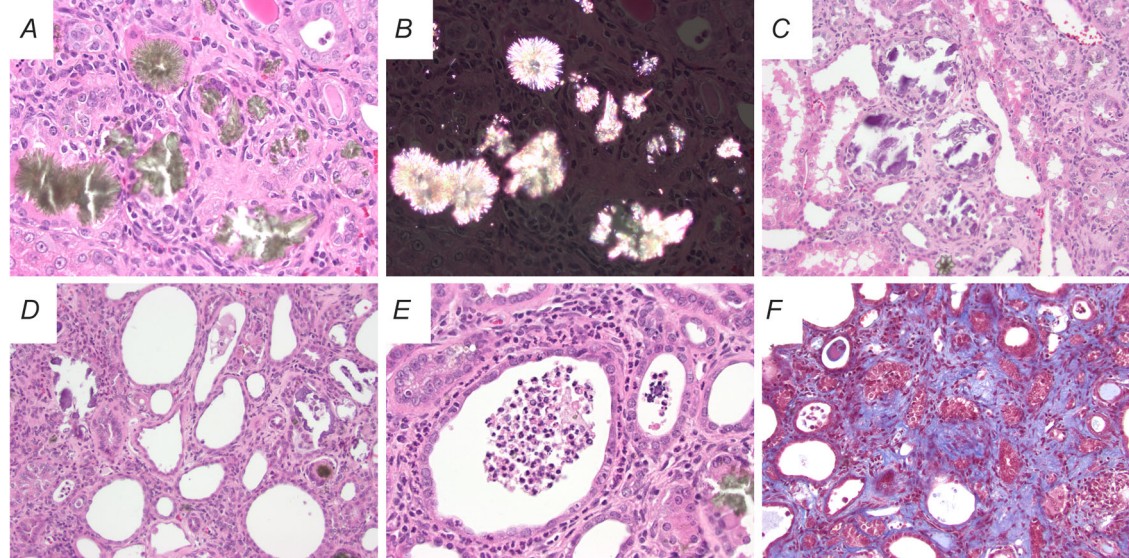

**Figure A4. Histological findings in adenine-diet-induced chronic kidney disease (_ADI_CKD) animals**
*A*, abundant 2,8-dihyroxyadenine crystals appear slightly brownish grey on haematoxylin and eosin (H&E) stain and (*B*) are birefringent under polarized light (400×). *C*, numerous large pale purple calcium phosphate crystals are seen on H&E stain (200×). Note the adjacent hypertrophic proximal tubules showing tubular injury. *D*, frequent tubules demonstrate microcystic dilatation, likely due to obstruction (H&E; 200×). *E*, scattered tubules contain luminal microabscesses with neutrophils and cell debris, likely indicating acute pyelonephritis (H&E; 400×). *F*, trichrome stain showing interstitial widening due to fibrosis (200×).

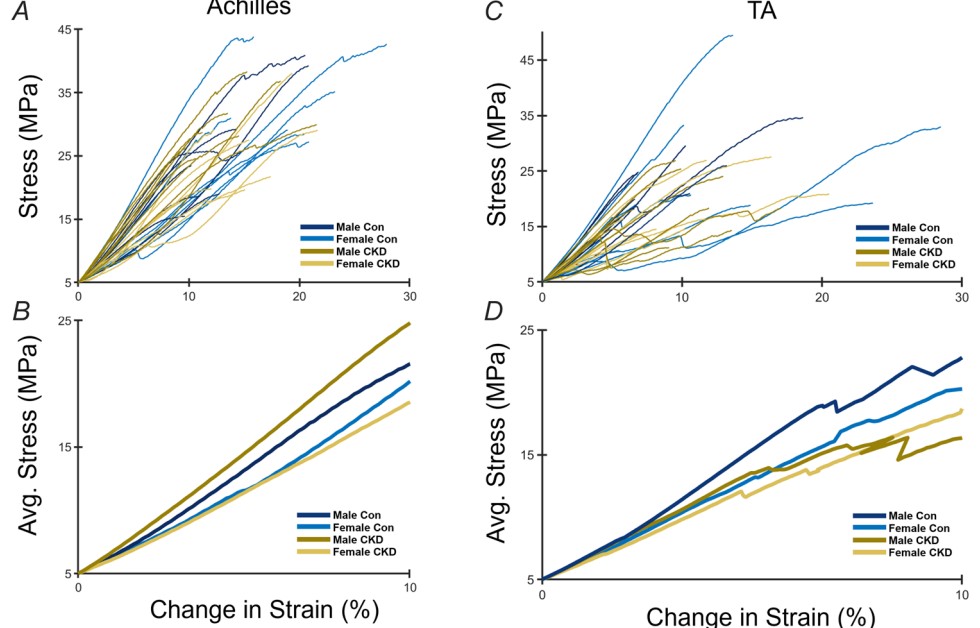

**Figure A5. Stress-strain curves of tendons normalized to strain at 5 MPa**
*A* and *C*, stress–strain curves of individual Achilles and tibialis anterior (TA) tendons from all four experiment groups (*n* = 8 per group). *B* and *D*, average stress–strain curves for each group, zoomed into highlight the typical linear region. To align traces for plotting and averaging strain was re-zeroed *post hoc* from the point at which stress equalled 5 MPa. C, control animals; CKD, animals with adenine-diet-induced chronic kidney disease.

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

## Additional information

### Data availability statement

All data points are shown in figures, and complete western blots are included in the appendix. The authors agreed to provide any and all raw data upon reasonable request.

### Competing interests

All authors declare no competing interests

### Author contributions

C.M.T.H., B.R. and K.B. contributed to project design. C.M.T.H., N.K.G., B.O., S.E.B., M.G. and K.Y.J. completed data acquisition and analysis. All authors contributed to the interpretation of the work and to drafting and revising it critically for important intellectual content. All authors approved the final version of the manuscript and agreed to be accountable for all aspects of the work in ensuring that questions related to the accuracy or integrity of any part of the work are appropriately investigated and resolved; and all persons designated as authors qualify for authorship, and all those who qualify for authorship are listed.

### Funding

This work was supported by the Peretz Family Fund for Connective Tissue Research. C.M.T.H. was supported by the National Institute of Diabetes and Digestive and Kidney Diseases U2C/TL1 LAUNCH programme (DK139565). N.K.G. was supported by the UC Davis T32 in Pharmacology (T32 GM144303). Kidney histology was paid for through B.R.'s Paul F. Gulyassy endowment.

### Acknowledgements

We sincerely thank the funding providers, all members of the Baar lab and the animal care services for making this work possible.

## Keywords

adenine, bone–mineral disorder, mechanics, muscle, rupture

## Supporting information

Additional supporting information can be found online in the Supporting Information section at the end of the HTML view of the article. Supporting information files available:

**Peer Review History**

