## [Peer Review History · The Journal of Physiology]

Kidney disease impairs tendon function in rats

Chris MT Hayden, Natalie K. Gilmore, Benjamin Osipov, Sarah E Brashear, Marc Gorge, Kuang-Yu Jen, Lucas Robert Smith, Blaine Christiansen, Baback Roshanravan, and Keith Baar

DOI: 10.1113/JP289753

Corresponding author(s): Keith Baar (kbaar@ucdavis.edu)

Review Timeline:	Submission Date:	22-Jul-2025
	Editorial Decision:	06-Oct-2025
	Revision Received:	28-Oct-2025
	Accepted:	03-Dec-2025

Senior Editor: Bettina Mittendorfer

Reviewing Editor: Hirotaka Iijima

Transaction Report:

Dear Dr Baar,

Re: JP-RP-2025-289753 "**Kidney disease impairs tendon function in rats**" by Chiris MT Hayden, Natalie K. Gilmore, Benjamin Osipov, Sarah E Brashear, Marc Gorge, Kuang-Yu Jen, Lucas Robert Smith, Blaine Christiansen, Baback Roshavan, and Keith Baar

Thank you for submitting your manuscript to The Journal of Physiology. It has been assessed by a Reviewing Editor and by 1 expert referee and we are pleased to tell you that it is potentially acceptable for publication following satisfactory major revision.

REVISION CHECKLIST:

We look forward to receiving your revised submission.

Yours sincerely,

Bettina Mittendorfer
Senior Editor
The Journal of Physiology

REQUIRED ITEMS

- Include a Key Points list in the article itself, before the Abstract.

- Author photo and profile. First or joint first authors are asked to provide a short biography (no more than 100 words for one author or 150 words in total for joint first authors) and a portrait photograph. These should be uploaded and clearly labelled together in a Word document with the revised version of the manuscript. See Information for Authors for further details.

- The reference list must be in alphabetical order, rather than numbered, to comply with our Journal format.

- Your manuscript must include a complete Additional Information section, including competing interests; funding; author contributions and acknowledgements.

- Please upload separate high-quality figure files via the submission form.

- You must upload original, uncropped western blot/gel images (including controls) if they are not included in the manuscript. This is to confirm that no inappropriate, unethical or misleading image manipulation has occurred. These should be uploaded as 'Supporting information for review process only'. Please label/highlight the original gels so that we can clearly see which sections/lanes have been used in the manuscript figures. For more information, see: <https://physoc.onlinelibrary.wiley.com/hub/journal-policies#imagmanip>.

- A Data Availability Statement is required for all papers reporting original data. This must be in the Additional Information section of the manuscript itself. It must have the paragraph heading 'Data Availability Statement'. All data supporting the results in the paper must be either: in the paper itself; uploaded as Supporting Information for Online Publication; or archived in an appropriate public repository. The statement needs to describe the availability or the absence of shared data. Authors must include in their statement: a link to the repository they have used, or a statement that it is available as Supporting Information; reference the data in the appropriate sections(s) of their manuscript; and cite the data they have shared in the References section. Whenever possible, the scripts and other artefacts used to generate the analyses presented in the paper should also be publicly archived. If sharing data compromises ethical standards or legal requirements then authors are not expected to share it, but must note this in their statement. For more information, see our Statistics Policy.

- Please include an Abstract Figure file, as well as the Figure Legend text within the main article file. The Abstract Figure is a piece of artwork designed to give readers an immediate understanding of the research and should summarise the main conclusions. If possible, the image should be easily 'readable' from left to right or top to bottom. It should show the physiological relevance of the manuscript so readers can assess the importance and content of its findings. Abstract Figures should not merely recapitulate other figures in the manuscript. Please try to keep the diagram as simple as possible and without superfluous information that may distract from the main conclusion(s). Abstract Figures must be provided by authors no later than the revised manuscript stage and should be uploaded as a separate file during online submission labelled as File Type 'Abstract Figure'. Please also ensure that you include the figure legend in the main article file. All Abstract Figures should be created using BioRender. Authors should use The Journal's premium BioRender account to export high-resolution images. Details on how to use and access the premium account are included as part of this email.

EDITOR COMMENTS

Reviewing Editor:

This study explored the effects of CKD on tendon, muscle, and bone, which is an underreported area of research. Reviewer #1 highlighted that the differences in tendon mechanics are modest, stress-strain responses should be fully shown, and histological assessment is needed, while also noting that muscle and bone findings could be emphasized more.

REFEREE COMMENTS

Referee #1:

The manuscript by Hayden et al addresses a potentially interesting, underreported effect of kidney disease on tendon function. The conclusion that there is a difference in tendons in female rats is weakened by the reporting that the induced CKD was more profound in males (and based on the literature, rupture appears to be more prevalent in the human male population - citation 17). The minor decrease in failure strength and no other changes in material properties does not provide a compelling argument that CKD leads to changes in tendon function in this model system. There does appear to be new data regarding the effects of CKD skeletal muscle and bone and these should be emphasized more.

Abstract

- The results for muscle and bone need to be described in the abstract.

Introduction

- It needs to be described where the spontaneous ruptures have been reported to occur in humans (e.g. tendon midsubstance, enthesis or myotendinous junction), whether it occurs predominantly in load bearing or positional tendons, and if it is more prevalent in male vs female tendons. The activities that lead to spontaneous ruptures in people with CKD also need to be defined. Does this happen with normal ambulation or only under exercise or increased loading? All of these topics then need to be brought up in the discussion in the context of the results.

Methods

- Figure 1: number of biological replicates and error bars need to be defined in the figure legend. Error bars need to be shown for all data points for weight and food intake.
- Young's modulus not a material property that viscoelastic tissues have. The stress-strain responses of the tendons should be shown so the reader and reviewer can assess how the material properties vary. Comparing the slopes of stress-strain curves is not useful unless the slopes were all measured at the same strain. Where the slope was measured needs to be more clearly stated; "calculated as the steepest slope (calculated using a least-squares fit) during the initial linear portion of the ... stress-strain curves," is vague. Accurate reporting of the material properties may actually show a significant difference in the tissues.

Results

- Figure 2: It is stated that there was "an increase in collagen deposition, 2,8-dihydroxyadenine crystals, calcium phosphate crystals, neutrophil infiltration, tubule dilation, and proximal tubule hypertrophy in the adenine-diet fed animals." These need to all be indicated on the figures to help the reader interpret the results.
- Figure 4: Stats need to be included for PDH and ETC graphs.
- Figure 5: For a conclusion to be made about the failure stress, where the tendons failed needs to be stated and they should predominantly fail in the midsubstance. If the tendons are simply pulling out of the grips in the region where the muscle was removed, the differences could be an artifact of the effect of CKD on the residual muscle ECM architecture, or differences in tissue size as a function of sex.
- Histology needs to be carried out to investigate differences in tendon organization (similar to what was done with the kidney in figure 2).

Discussion

- Since it was described in the introduction that the adenine diet induced model is well characterized in rodents including results for bone, more description is needed to indicate to the reader what new/different scientific information is added by the data shown in figures 2 and 3 that has not been published already.
- It is unclear how the males can show a more profound difference in the development of CKD yet show little difference in material properties of the tendon. Furthermore, it appears from the literature that mainly load bearing tendons are the ones that fail, yet it is the TA that shows a significant effect of diet. These discrepancies need to be discussed.
- Why there was a discrepancy in kidney weight as a function of left and right sides should be discussed

END OF COMMENTS

We want to thank the reviewer for their comments and positive feedback on the work. We think that the comments will greatly improve the readability of the work. Below, we reply to each of the individual points raised and how we have addressed this in the revised manuscript. Below, we highlight our replies in blue text following a ">>" notation.

Reviewing Editor:

This study explored the effects of CKD on tendon, muscle, and bone, which is an underreported area of research. Reviewer #1 highlighted that the differences in tendon mechanics are modest, stress-strain responses should be fully shown, and histological assessment is needed, while also noting that muscle and bone findings could be emphasized more.

>>Thank you. We have tried to address each of the comments from the reviewer.

Referee #1:

The manuscript by Hayden et al addresses a potentially interesting, underreported effect of kidney disease on tendon function. The conclusion that there is a difference in tendons in female rats is weakened by the reporting that the induced CKD was more profound in males (and based on the literature, rupture appears to be more prevalent in the human male population - citation 17). The minor decrease in failure strength and no other changes in material properties does not provide a compelling argument that CKD leads to changes in tendon function in this model system. There does appear to be new data regarding the effects of CKD skeletal muscle and bone and these should be emphasized more.

Abstract

- The results for muscle and bone need to be described in the abstract.

>> The following has been added to the abstract (which is constrained by a 250 word limit):

“Plantar flexor strength was also 13% lower in _{ADI}CKD ($p=0.021$), and femur yield force was decreased by 41% in male _{ADI}CKD alone ($p<0.001$).”

Introduction

- It needs to be described where the spontaneous ruptures have been reported to occur in humans (e.g. tendon midsubstance, enthesis or myotendinous junction), whether it occurs predominantly in load bearing or positional tendons, and if it is more prevalent in male vs female tendons. The activities that lead to spontaneous ruptures in people with CKD also need to be defined. Does this happen with normal ambulation or only under exercise or increased loading? All of these topics then need to be brought up in the discussion in the context of the results.

>> The following has been added to the intro:

“Tendons, for instance, have received relatively little attention in the CKD literature, which is concerning, as there are more than 60 years of case studies describing spontaneous (non-traumatic) tendon ruptures in patients with CKD (Wilson, 1957; Preston, 1972; Shah, 2002; Tsourvakas *et al.*, 2004; Üreten *et al.*, 2008; Basic-Jukic *et al.*, 2009; Zaidenberg *et al.*, 2015; Taşoğlu *et al.*, 2016; Zhang *et al.*, 2020; Fakru *et al.*, 2021; Allata *et al.*, 2023). These reported ruptures occur in both load bearing (Achilles and quadriceps) and non-weight bearing tendons (biceps brachii, triceps), often during normal daily activities such as walking downstairs. The site of rupture varies across case reports and may occur at the osteotendinous junction, myotendinous junction, or the tendon mid-substance (Lotem *et al.*, 1978; Shah, 2002). “

And this has been added to the discussion:

“The only large scale analysis to date of tendon rupture in CKD (Humbyrd *et al.*, 2018), suggests that rupture is less common in men than in women (adjusted incidence rate ratio, aIRR=0.75). Although our study contradicts this somewhat, we believe this is due to the negative effects of obesity and inactivity in the male control group rather than a less pronounced effect of CKD. Consistent with this hypothesis, Humbyrd and colleagues also found that rupture risk increased with increasing BMI (reference BMI 0-24.99, BMI=25-29.99: aIRR=1.25, BMI=30-39.99: aIRR=1.56, BMI≥40: aIRR=1.81). Importantly, the Humbyrd study included only end-stage renal disease patients who were on dialysis or had already received a kidney transplant, which is different than our model.”

Methods

- Figure 1: number of biological replicates and error bars need to be defined in the figure legend. Error bars need to be shown for all data points for weight and food intake.

>>The figure heading has been updated to the following:

Figure 1. Study overview and time course of bodyweight, food intake, and estimated adenine dosage in rats fed control of adenine containing diets.

Performance testing consisted of treadmill running and electrically stimulated muscle contraction for assessment of strength and endurance of plantarflexors. Standard deviation is omitted where bars are smaller than symbol size (n=8 per group). Standard deviation could not be accurately calculated for adenine dosage and as such should be considered an estimation. ADICKD, adenine-diet-induced chronic kidney disease.

- Young's modulus is not a material property that viscoelastic tissues have.

>>Although tendons are viscoelastic, and therefore strain rate sensitive, measurement of modulus in tendon is common – including number of publications in this journal (<https://doi.org/10.1113/JP286609>, <https://doi.org/10.14814/phy2.70515>, <https://doi.org/10.14814/phy2.14267>). Our testing consisted of a relatively slow strain rate (0.25mm/s) which has been suggested in the ex vivo tendon testing guidelines paper to reduce the impact of tendon viscosity, sometimes referred to as quasi-static testing (<https://doi.org/10.14814/phy2.14267>). However, to satisfy the reviewer, we have simply switched the measure to modulus and defined this in the methods as the maximal slope of the stress-strain curve.

- The stress-strain responses of the tendons should be shown so the reader and reviewer can assess how the material properties vary.

>>Stress strain curves (normalized to strain at 5MPa), have been added as an Appendix figure. This includes every individual trace as well as group averages.

- Comparing the slopes of stress-strain curves is not useful unless the slopes were all measured at the same strain. Where the slope was measured needs to be more clearly stated; "calculated as the steepest slope (calculated using a least-squares fit) during the initial linear portion of the ... stress-strain curves," is vague. Accurate reporting of the material properties may actually show a significant difference in the tissues.

>> While not measured at the exact same strain for each tendon, the comparison we make is of the maximum stress/strain slope in response to a standardized strain rate. The viscous component of the stress/strain curve is dependent on the strain rate not magnitude and should therefore be relatively consistent. The determination of maximum modulus, as used by other leaders in the field (<https://doi.org/10.14814/phy2.14267>), also helps standardize across tissues representing the greatest capacity of the tissue to resist elongation. The modulus we report here would thus be best fully defined as the maximum apparent Young's modulus. The wording has been changed in the manuscript for clarification.

Results

- Figure 2: It is stated that there was "an increase in collagen deposition, 2,8-dihydroxyadenine crystals, calcium phosphate crystals, neutrophil infiltration, tubule

dilation, and proximal tubule hypertrophy in the adenine-diet fed animals." These need to all be indicated on the figures to help the reader interpret the results.

>>The following figure and caption highlighting these results has been added to the Appendix:

Figure #: Histologic findings in adenine-diet fed animals. A) Abundant 2,8-dihydroxyadenine crystals appear slightly brownish-grey on H&E stain and B) are birefringent under polarized light (400X). C) Numerous large pale purple calcium phosphate crystals are seen on H&E stain (200X). Note the adjacent hypertrophic proximal tubules showing tubular injury. D) Frequent tubules demonstrate microcystic dilation, likely due to obstruction (H&E; 200X). E) Scattered tubules contain luminal microabscesses with neutrophils and cell debris, likely indicating acute pyelonephritis (H&E; 400X). F) Trichrome stain showing interstitial widening due to fibrosis (200X).

- Figure 4: Stats need to be included for PDH and ETC graphs.

>>P-values have been added to these plots. The following description has been added to the figure legend: "Protein levels were compared between diets separately for each sex using an unpaired t-test". This has also been highlighted in the *Statistics* section of the methods: "Unpaired t-tests were used to compare protein levels (determined by Western blot) across diet interventions for each sex separately, to ensure comparisons were made only within a single blot."

- Figure 5: For a conclusion to be made about the failure stress, where the tendons failed needs to be stated and they should predominantly fail in the midsubstance. If the tendons are simply pulling out of the grips in the region where the muscle was removed,

the differences could be an artifact of the effect of CKD on the residual muscle ECM architecture, or differences in tissue size as a function of sex.

>>We agree with the reviewer that failure stress needs to be properly defined to make it clear what we are measuring. Because our test proceeds until an 80% drop in force occurs, our tendons do not completely separate, and the exact failure point is sometimes not obvious. In response to peer review comments on other projects we have completed and published we have switched from the term “ultimate tensile stress” to failure stress to reflect the fact that tendons may not always fail at the mid-substance. The methods section has been edited as follows:

“Samples were immersed in 0.9% PBS within the BioPuls bath held at 37°C, preconditioned using 10 cycles (0.10–0.25 N, 0.25 mm/s), and then elongated at 0.25 mm/s until failure – defined as an 80% reduction in force. Force and displacement were recorded throughout. From this data, the maximal tensile load (MTL; N) and failure stress (MTL divided by CSA; MPa) were determined as the peak force and peak stress obtained during the test, respectively. When not visually obvious, failure was confirmed by gently passing a pair of forceps through the remaining tendon tissue.”

- Histology needs to be carried out to investigate differences in tendon organization (similar to what was done with the kidney in figure 2).

>>We agree with the reviewer that histology is an important next step to determine the etiology of the tendon dysfunction we observed. In this initial study, we chose to prioritize obtaining measures of mechanical function, collagen content (hydroxyproline assay), and matrix stability (differential scanning calorimetry) which are all incompatible with histological analysis. Unfortunately, this means we have no remaining tissue for histology; however, we do intend to perform both traditional histology and second-harmonic generation imaging in follow-up work.

Discussion

- Since it was described in the introduction that the adenine diet induced model is well characterized in rodents including results for bone, more description is needed to indicate to the reader what new/different scientific information is added by the data shown in figures 2 and 3 that has not been published already.

>>The following has been added to the discussion and has also been highlighted in the “Key Points” section added to the beginning of the manuscript:

“This is the first study known to the authors to combine both functional and structural analysis of these tissues into one paper. As muscle-bone crosstalk in CKD has recently become a topic of interest (Leal *et al.*, 2021; Wong & McMahon, 2023), the data presented here support the use of this model to mechanistically explore this area.

Furthermore, the analysis of tendon tissue is completely novel and together with the muscle and bone data creates a platform for comprehensive evaluation of the effects of the CKD on musculoskeletal function.”

- It is unclear how the males can show a more profound difference in the development of CKD yet show little difference in material properties of the tendon. Furthermore, it appears from the literature that mainly load bearing tendons are the ones that fail, yet it is the TA that shows a significant effect of diet. These discrepancies need to be discussed.

>>The following has been added to the discussion to address these points:

Regarding CKD dose response:

“Amongst our three musculoskeletal tissues, it may be noted that bone is the only one where the greater burden of CKD in males appeared to result in more severe pathology. The lack of apparent dose response in muscle and tendon to CKD might suggest that the pathobiology of kidney disease occurs at lower levels of disease or that the metabolic impact of hyperphagia and the higher BMI was more detrimental to muscle and tendon than bone. Consistent with this hypothesis, we have previously demonstrated that diet-induced obesity negatively affects muscle and tendon function in mice (PMID: 38882396). The mechanisms that lead to bone dysfunction in CKD may differ from those driving the loss of muscle and tendon function. In fact, a surprising finding of Humbyrd and colleagues was that amongst individuals with ESRD, those with osteoporosis were far less likely to spontaneously rupture their Achilles than those without (aIRR=0.14)(Humbyrd *et al.*, 2018).”

Regarding location of tendon pathology:

“Spontaneous tendon ruptures in CKD have been reported in many different tendons of the upper and lower extremities, often occurring in load bearing tendons of the knee and ankle - although there is at least one report of a torn medial obliquus abdominus internus (Basic-Jukic *et al.*, 2009). Our finding that the TA tendon was more consistently affected by CKD than the load bearing Achilles, should be evaluated in context with two thoughts. One, tendon loading within healthy bounds results in positive tissue adaptation that may counteract the negative effect of disease (e.g., the benefits of exercise)(Chen *et al.*, 2020). Two, although adult tendons are relatively stable (i.e., low protein turnover) and may need external stimulus to initiate pathology, these animals developed kidney disease while their musculoskeletal system was still growing and maturing. Thus, in this model the negative effects of CKD on tendons may be evident in more locations as CKD may impair the normal tendon development. Nevertheless, the finding of worse material properties in TA tendon is good evidence for a direct systemic

effect of CKD on tendon tissue, as opposed to confounding effects of other CKD symptoms such as sedentarism. “

- Why there was a discrepancy in kidney weight as a function of left and right sides should be discussed

>>The following has been added to the discussion to address these points:

“As for the interesting finding that in _{ADI}CKD the left kidney is consistently significantly larger than the right, a potential explanation is that this occurs due to differences in renal blood flow. The left kidney has a shorter artery and longer vein, meaning there is less resistance to afferent flow and more resistance to efferent flow, while the right kidney has the opposite. This could potentially lead to differences in glomerular pressure and therefore hypertrophy, as our histology indicates obstruction induced dilation. However, this is highly speculative and at the moment the authors have no definitive evidence for this.

Dear Dr Baar,

Re: JP-RP-2025-289753R1 "**Kidney disease impairs tendon function in rats**" by Chris MT Hayden, Natalie K. Gilmore, Benjamin Osipov, Sarah E Brashear, Marc Gorge, Kuang-Yu Jen, Lucas Robert Smith, Blaine Christiansen, Baback Roshanravan, and Keith Baar

We are pleased to tell you that your paper has been accepted for publication in The Journal of Physiology.

Yours sincerely,

Bettina Mittendorfer
Senior Editor
The Journal of Physiology

IMPORTANT POINTS TO NOTE FOLLOWING ACCEPTANCE OF YOUR PAPER:

- **IMPORTANT NOTICE ABOUT OPEN ACCESS:** To assist authors whose funding agencies mandate immediate public access to published research findings, The Journal of Physiology allows authors to pay an Open Access (OA) fee to have their papers made freely available immediately on publication.

- You can help your research get the attention it deserves! Check out Wiley's free Promotion Guide for best-practice recommendations for promoting your work at: www.wileyauthors.com/eoo/guide. You can learn more about Wiley Editing Services which offers professional video, design, and writing services to create shareable video abstracts, infographics, conference posters, lay summaries, and research news stories for your research at: www.wileyauthors.com/eoo/promotion.

- If you would like to receive our 'Research Roundup', a monthly newsletter highlighting the cutting-edge research published in The Physiological Society's family of journals (The Journal of Physiology, Experimental Physiology, Physiological Reports, The Journal of Nutritional Physiology and The Journal of Precision Medicine: Health and Disease), please click this link, fill in your name and email address and select 'Research Roundup': <https://www.physoc.org/journals-and-media/membernews>

EDITOR COMMENTS

Reviewing Editor:

The careful revision and detailed responses to the reviewers have strengthened the manuscript and resulted in a more robust conclusion.

REFeree COMMENTS

Referee #1:

The revisions and additional data strengthen the study.